



**Ross scheme, Newton-Raphson iterative methods and time-stepping strategies for solving the mixed-form of Richards' equation**

Hassane[1,2] Mamadou Maina F. Z., Ackerer[1,]* P.

[1]Laboratoire d'Hydrologie et de Géochimie de Strasbourg,
Univ. Strasbourg/EOST - CNRS
1 rue Blessig, 67084 Strasbourg,
France

[2]CEA-Laboratoire de Modélisation des Transferts dans l'Environnement,
Bât. 225, F-13108 Saint Paul lez Durance cedex,
France

*Corresponding author: ackerer@unistra.fr

**ABSTRACT**

The solution of the mathematical model for flow in variably saturated porous media described
by Richards equation (RE) is subject to heavy numerical difficulties due to its highly non-
linear properties and remains very challenging. Two different algorithms are used in this work
to solve the mixed-form of RE: the traditional iterative algorithm and a time-adaptive
algorithm consisting of changing the time step magnitude within the iteration procedure while
the state variable is kept constant. The Ross method is an example of this type of scheme, and
we show that it is equivalent to the Newton-Raphson method with a time-adaptive algorithm.
Both algorithms are coupled to different time stepping strategies: the standard heuristic
approach based on the number of iterations and two strategies based on the time truncation
error or on the change of water saturation. Three different test cases are used to evaluate the
efficiency of these algorithms.



The numerical results highlight the necessity of implementing two types of errors: the
iterative convergence error (maximum difference of the state variable between two iterations)
and an estimate of the time truncation errors. The algorithms using these two types of errors
together were found to be the most efficient when highly accurate results are required.
**Key words**: Unsaturated flow, Newton-Raphson, Time stepping





**1. Introduction**
Water movement in soils is one of the key processes in the water cycle since it contributes to
the renewal of groundwater resources through recharge, to vegetation growth through
transpiration, to soil fertility through salinization/alteration and to atmospheric humidity
through evaporation and transpiration. Water movement is usually modeled using the
Richards equation (Richards, 1931), which is now commonly adopted for many studies in soil
science and/or hydrology, including the use of physically based hydrological models applied
to large-scale catchments and for long time simulations (e.g., for climate change studies).
However, this equation is highly nonlinear and despite numerous efforts over the last 40
years, its numerical solution requires much computational time.
Assuming a rigid solid matrix, the Richards equation (RE) is given by,

$$\begin{cases} \dfrac{\partial \theta}{\partial t} + S_w s_0 \dfrac{\partial \psi}{\partial t} + \nabla \cdot \mathbf{q} = f \\ \mathbf{q} = -k_r(\psi)\mathbf{K}\left[\nabla \psi + \nabla z\right] \end{cases} \tag{1}$$

where $\theta$ is the volumetric water content [$L^3/L^3$], $S_w$ is the water saturation [-], $s_0$ is the specific
storage coefficient [$L^{-1}$], $\psi$ is the pressure head [L], $\mathbf{q}$ is the water flux based on the extended
Darcy's law [L/T], t is the time [T], z is the vertical coordinate (positive upward) [L], f is the
sink/source term [$T^{-1}$], $\mathbf{K}$ is the saturated hydraulic conductivity tensor [L/T] and $k_r(\psi)$ is the
relative hydraulic conductivity [-]. The model includes initial and boundary conditions of the
Dirichlet (prescribed pressure head) or Neumann (prescribed flux) type.
Equation (1) is also called the mixed form of RE. Two alternative formulations exist for RE.
The pressure form is defined by:

$$\begin{cases} \left[C(\psi) + S_w s_0\right]\dfrac{\partial \psi}{\partial t} + \nabla \cdot \mathbf{q} = f \\ \mathbf{q} = -k_r(\psi)\mathbf{K}\left[\nabla \psi + \nabla z\right] \end{cases} \tag{2}$$




where $C(\psi) = \dfrac{\partial \theta}{\partial \psi}$ is the specific moisture capacity [L$^{-1}$], and the soil moisture form that is
restricted to unsaturated conditions is defined by:

$$
\begin{cases}
\dfrac{\partial \theta}{\partial t} + \nabla \cdot \mathbf{q} = f \\
\mathbf{q} = -\left( \mathbf{D}(\theta) \nabla \theta + k_r(\theta) \mathbf{K} \nabla z \right)
\end{cases}
\tag{3}
$$

where $\mathbf{D}(\theta) = k_r(\theta) \mathbf{K} \dfrac{d\psi}{d\theta}$ is the pore water diffusivity [L$^2$/T].
Constitutive relations are required to solve RE. For the pressure-water content relationship,
the most common model is the Van Genuchten model (van Genuchten, 1980):

$$
S_w(\psi) = \frac{\theta(\psi) - \theta_r}{\theta_s - \theta_r} =
\begin{cases}
\left( 1 + |\alpha \psi|^\eta \right)^{-m} & \psi < 0 \\
1 & \psi \geq 0
\end{cases}
\tag{4}
$$

where $m = 1 - 1/\eta$, $S_w$ is the effective saturation, $\theta_r$ and $\theta_s$ are the residual and saturated
volumetric water content respectively, $\alpha$ and $\eta$ are experimentally estimated coefficients.
This model is usually associated with Mualem model (Mualem, 1976) for the relative
permeability of the aqueous phase:
$$
k_r(S_w) =
\begin{cases}
S_w^{1/2} \left[ 1 - \left( 1 - S_w^{1/m} \right)^m \right]^2 & \psi < 0 \\
1.0 & \psi \geq 0
\end{cases}
\tag{5}
$$

A summary of the most popular relations can be found in Belfort et al. (2013).
Due to the strong heterogeneities of the unsaturated zone and nonlinearities in the constitutive
relations (Eq. (4) and (5)), analytical solution of RE does not exist except in special cases
(Celia et al., 1990; van Dam and Feddes, 2000). Therefore, numerical methods such as finite
difference (Feddes et al., 1988; Romano et al., 1998; van Dam and Feddes, 2000), finite



element (Gottardi and Venutelli, 2001), and mixed finite element (Bause and Knabner, 2004;
Bergamaschi and Putti, 1999; Fahs et al., 2009; Farthing et al., 2003) are used to solve RE.
Iterative methods based on the Picard (fixed point) or Newton-Raphson approach (Lehmann
and Ackerer, 1998; Paniconi and Putti, 1994) are the most popular techniques for solving this
highly nonlinear equation. Alternative iterative methods are based on transform formulations
(Crevoisier et al., 2009; Ross and Bristow, 1990; Williams et al., 2000; Zha et al., 2013) or
the method of lines (Fahs et al., 2009; Matthews et al., 2004; Miller et al., 1998; Tocci et al.,
1997). Additionally, very few non-iterative schemes have been developed (Kavetski and
Binning, 2004, 2002a; Paniconi et al., 1991).

Despite the many existing numerical methods, solution of the RE is still a challenging
research topic with many remaining questions about reduction of the computational time,
treatment of nonlinearities, and improvement of the accuracy of these methods for difficult
problems such as infiltration in very dry soils (Diersch and Perrochet, 1999; Forsyth et al.,
1995; R. G. Hills, 1989).
In this study, we analyzed the performance of different algorithms based on the Newton-
Raphson method since the classical Picard scheme has been found less efficient (Lehmann
and Ackerer, 1998). Applied to the soil moisture form of the RE equation, we demonstrate
that the recently developed Ross method (Ross, 2003; Crevoisier et al., 2009; Zha et al., 2013)
is equivalent to Newton-Raphson method (section 2). A detailed presentation of the Newton-
Raphson method applied to the mixed form or RE is given in section 3. The standard Newton-
Raphson algorithm is based on the computation of the corresponding matrices in an iterative
way by updating the parameters until convergence. An alternative algorithm has been
suggested more recently where the parameters are kept unchanged within one time step and




the time step is adapted to reach convergence. This algorithm has been applied to the
pressure-based form of RE by Kavetski and Binning (2002a) and to the soil moisture form by
Crevoisier et al. (2009), Ross (2003), Zha et al. (2013). Although this algorithm is called "non
iterative" because the parameters are not updated during the calculation, iterations may be
necessary to adapt the magnitude of the time step. Therefore, in the following, we will refer to
the usual algorithm as "iterative" and to the alternative algorithm as "time-adaptive". To our
knowledge, this alternative algorithm has never been applied to the mixed form of RE.
Section 4 is dedicated to both algorithms and to the time stepping strategy used for solving
RE. Finally, in section 5, the numerical accuracy and robustness of the algorithms applied to
the mixed-form of RE are evaluated using three different test cases.

**2. The Ross method and the Newton-Raphson method**
The moisture-based formulation is applicable in unsaturated conditions only and is prone to
numerical difficulties in the case of heterogeneous soils, explaining the reduced attention
directed to this formulation. However, discontinuous water content can be handled by adapted
schemes and moisture-based formulation appears to be very accurate for initially dry
conditions (Zha et al., 2013, 2015).
Ross (2003) suggested a non-iterative formulation that has been recently extended to different
soil conditions (Crevoisier et al., 2009; Varado et al., 2006a) and to two and three dimensions
(Zha et al., 2013).
In its initial one-dimensional finite-volume formulation and for a volume (cell) $i$, the Ross
method (Ross, 2003) is based on the following set of equations:
$$\frac{\Delta z}{\Delta t}\left(\theta_i^{n+1} - \theta_i^n\right) = \frac{\Delta z}{\Delta t}\left(\theta_{s,i} - \theta_{r,i}\right)\left(S_i^{n+1} - S_i^n\right) = q_-^\sigma - q_+^\sigma \qquad (6)$$
with:
$$\begin{cases} q_+^\sigma = q_+^n + \sigma\left[\left(\frac{\partial q_i^n}{\partial S_i^n}\right)\left(S_i^{n+1} - S_i^n\right) + \left(\frac{\partial q_i^n}{\partial S_{i+1}^n}\right)\left(S_{i+1}^{n+1} - S_{i+1}^n\right)\right] \\ q_-^\sigma = q_-^n + \sigma\left[\left(\frac{\partial q_i^n}{\partial S_i^n}\right)\left(S_i^{n+1} - S_i^n\right) + \left(\frac{\partial q_i^n}{\partial S_{i-1}^n}\right)\left(S_{i-1}^{n+1} - S_{i-1}^n\right)\right] \end{cases} \qquad (7)$$



where $S_i^{n+1}$ is the water saturation at cell/node $i$ at time $(n+1)$, $q_-^\sigma$ (resp. $q_+^\sigma$) is the water flux
between cell $i$ and $(i\text{-}1)$ (resp. $i+1$) at time $t = t^n + \sigma\,\Delta t$, $\sigma \in [0,1]$ and $\Delta z$ is the size of the
cell $i$. $\theta_{s,i}$ is the saturated water content and $\theta_{r,i}$ is the residual water content. For simplicity,
we assume here that all cells are of the same size.
The previous mass balance equation (6) leads to the following equation for cell $i$:

$$
\begin{aligned}
-\left(\frac{\partial q_-^n}{\partial S_{i-1}^n}\right)\left(S_{i-1}^{n+1}-S_{i-1}^n\right) &+ \left[\frac{\Delta z}{\sigma\Delta t}\left(\theta_{s,i}-\theta_{r,i}\right)-\left(\left(\frac{\partial q_-^n}{\partial S_i^n}\right)-\left(\frac{\partial q_+^n}{\partial S_i^n}\right)\right)\right]\left(S_i^{n+1}-S_i^n\right) \\
&+\left(\frac{\partial q_+^n}{\partial S_{i+1}^n}\right)\left(S_{i+1}^{n+1}-S_{i+1}^n\right) = q_-^n - q_+^n
\end{aligned}
\tag{8}
$$

The Newton-Raphson method was initially developed as a root-finding algorithm of an
arbitrary equation that has been generalized for solving a system of non-linear equations.
Applied to the soil moisture form of the RE and using an implicit scheme, the NR consists in
defining a residual based on the mass balance equation (Eq. (6)) at iteration $k$ for time step
$n+1$ and for cell $i$ written as:

$$
R_i^{n+1,k} = \frac{\Delta z}{\Delta t}\left(\theta_{s,i}-\theta_{r,i}\right)\left(S_i^{n+1,k}-S_i^n\right)+q_+^{n+1,k}-q_-^{n+1,k}
\tag{9}
$$

where $R_i^{n+1,k}$ is called the residual.
The NR consists in computing the solution at iteration $k+1$ by estimating the residual of the
next iteration $R_i^{n+1,k+1}$ using a first order Taylor development and setting it equal to zero as:

$$
\frac{R_i^{n+1,k}}{\partial S^{n+1,k}}\left(S_i^{n+1,k+1}-S_i^{n+1,k}\right)+R_i^{n+1,k}=0
\tag{10}
$$

The derivatives of this residual are:

$$
\begin{cases}
\dfrac{\partial R_i^{n+1,k}}{\partial S_{i-1}^{n+1,k}} = -\dfrac{\partial q_-^{n+1,k}}{\partial S_{i-1}^{n+1,k}} \\[2ex]
\dfrac{\partial R_i^{n+1,k}}{\partial S_i^{n+1,k}} = \dfrac{\Delta z}{\Delta t}\left(\theta_{s,i}-\theta_{r,i}\right)+\dfrac{\partial q_+^{n+1,k}}{\partial S_i^{n+1,k}}-\dfrac{\partial q_-^{n+1,k}}{\partial S_i^{n+1,k}} \\[2ex]
\dfrac{\partial R_i^{n+1,k}}{\partial S_{i+1}^{n+1,k}} = \dfrac{\partial q_+^{n+1,k}}{\partial S_{i+1}^{n+1,k}}
\end{cases}
\tag{11}
$$





which leads to the following set of linear equations:

$$-\frac{\partial q_-^{n+1,k}}{\partial S_{i-1}^{n+1,k}}\left(S_{i-1}^{n+1,k+1}-S_{i-1}^{n+1,k}\right)+\left[\frac{\Delta z}{\Delta t}\left(\theta_{s,i}-\theta_{r,i}\right)+\frac{\partial q_+^{n+1,k}}{\partial S_i^{n+1,k}}-\frac{\partial q_-^{n+1,k}}{\partial S_i^{n+1,k}}\right]\left(S_i^{n+1,k+1}-S_i^{n+1,k}\right)$$

$$+\frac{\partial q_+^{n+1,k}}{\partial S_{i+1}^{n+1,k}}\left(S_{i+1}^{n+1,k+1}-S_{i+1}^{n+1,k}\right)=\frac{\Delta z}{\Delta t}\left(\theta_{s,i}-\theta_{r,i}\right)\left(S_i^{n+1,k}-S_i^n\right)+q_+^{n+1,k}-q_-^{n+1,k}$$

(12)


For the first iteration, we have $S_i^{n+1,k+1}=S_i^{n+1}$ and $S_i^{n+1,k}=S_i^n$, and therefore :

$$-\frac{\partial q_-^n}{\partial S_{i-1}^n}\left(S_{i-1}^{n+1}-S_{i-1}^n\right)+\left[\frac{\Delta z}{\Delta t}\left(\theta_{s,i}-\theta_{r,i}\right)+\frac{\partial q_+^n}{\partial S_i^n}-\frac{\partial q_-^n}{\partial S_i^n}\right]\left(S_i^{n+1}-S_i^n\right)$$

$$+\frac{\partial q_+^n}{\partial S_{i+1}^n}\left(S_{i+1}^{n+1}-S_{i+1}^n\right)=q_+^{n,k}-q_-^{n,k}$$

(13)


Whatever the formulation of the fluxes q (as a function of the pressure or the water content,
expressed by Kirchhoff transform as in Ross (2003) or not), the implicit Ross method (eq. (8)
with $\sigma=1$) ) is equivalent to the first iteration of the Newton-Raphson method (eq. (13)).

## 3. Newton Raphson method for the mixed form Richards' equation

Because the pressure-based formulation does not ensure mass conservation - except for the
approximation provided by Rathfelder and Abriola (1994) - and due to the limitations of the
moisture-based formulation (see previous section), the mixed formulation has been widely
used since the work of Celia et al. (1990).
The mixed form of the Richards equation given by equation (1) is rewritten as:

$$\frac{\partial \theta}{\partial t}+S_w s_0 \frac{\partial \psi}{\partial t}=\nabla \cdot k_r(\psi)\mathbf{K}\left[\nabla \psi+\nabla z\right]+f$$

(14)

and is discretized by:

$$\mathbf{A}^{n+1,k}\boldsymbol{\psi}^{n+1,k+1}+\mathbf{B}^{n+1,k}\frac{\boldsymbol{\psi}^{n+1,k+1}-\boldsymbol{\psi}^n}{\Delta t^{n+1}}+\mathbf{E}\frac{\boldsymbol{\theta}^{n+1,k+1}-\boldsymbol{\theta}^n}{\Delta t^{n+1}}=\mathbf{F}^{n+1,k}$$

(15)



where **A** is the discretized form of the divergence term, **B** and **E** are the discretized forms of
the storage terms and **F** is the discretized form of the sink/source term and the boundary
conditions, $n$ is the time step and $k$ the iteration counter. $\Delta t^{n+1}$ is the time step magnitude
defined by $\Delta t^{n+1} = t^{n+1} - t^n$. Matrices **A, B, E** and vector **F** depend on the numerical scheme
used for the spatial discretization. The implicit scheme is applied for the spatial discretization.
For the Newton-Raphson method, the residual is defined now by:
$$\mathbf{R}(\boldsymbol{\psi}^{n+1,k}) = \mathbf{A}^{n+1,k}\boldsymbol{\psi}^{n+1,k} + \mathbf{B}^{n+1,k}\frac{\boldsymbol{\psi}^{n+1,k} - \boldsymbol{\psi}^n}{\Delta t^{n+1}} + \mathbf{E}\frac{\boldsymbol{\theta}^{n+1,k} - \boldsymbol{\theta}^n}{\Delta t^{n+1}} - \mathbf{F}^{n+1,k} \tag{16}$$

and its derivatives are:
$$\mathbf{R}'(\boldsymbol{\psi}^{n+1,k}) = \mathbf{A}^{n+1,k} + \frac{\partial \mathbf{A}^{n+1,k}}{\partial \boldsymbol{\psi}^{n+1,k}}\boldsymbol{\psi}^{n+1,k} + \frac{\mathbf{B}^{n+1,k}}{\Delta t^{n+1}} + \frac{\partial \mathbf{B}^{n+1,k}}{\partial \boldsymbol{\psi}^{n+1,k}}\frac{\boldsymbol{\psi}^{n+1,k} - \boldsymbol{\psi}^n}{\Delta t^{n+1}}$$
$$+ \frac{\mathbf{E}}{\Delta t^{n+1}}\frac{\partial \boldsymbol{\theta}^{n+1,k}}{\partial \boldsymbol{\psi}^{n+1,k}} - \frac{\partial \mathbf{F}^{n+1,k}}{\partial \boldsymbol{\psi}^{n+1,k}} \tag{17}$$

Looking for $\boldsymbol{\psi}^{n+1,k+1}$ such as $\mathbf{R}(\boldsymbol{\psi}^{n+1,k+1}) = 0$, the system to solve is similar to Eq. (10):
$$\mathbf{R}'(\boldsymbol{\psi}^{n+1,k})\Delta\boldsymbol{\psi}^{n+1,k+1} = -\mathbf{R}(\boldsymbol{\psi}^{n+1,k}) \tag{18}$$

with $\Delta\boldsymbol{\psi}^{n+1,k+1} = \boldsymbol{\psi}^{n+1,k+1} - \boldsymbol{\psi}^{n+1,k}$.

The NR formulation is also used for the non-iterative scheme by applying only one NR step
per time step, with $\boldsymbol{\psi}^{n+1} = \boldsymbol{\psi}^{n+1,1}$ where $\boldsymbol{\psi}^{n+1,0} = \boldsymbol{\psi}^n$ (Paniconi et al., 1991; Zha et al., 2015).

**4. Algorithms and time stepping strategy**
The usual algorithm used to solve RE consists in defining a time step that remains constant
and to iteratively compute the parameters and variables in the following way:
***For a given time step n***
- Define the time step length $\Delta t^{n+1}$ depending on the time stepping strategy.





-    Initialization of the iterative process by setting $\psi^{n+1,1} = \psi^n$.

*do k=1, maxit*

1.  Computation of the variable $\theta^{n+1,k}$, the parameter $\mathbf{K}^{n+1,k}$ and their derivatives

$\dfrac{d\theta^{n+1,k}}{d\psi^{n+1,k}}, \dfrac{\partial\mathbf{K}^{n+1,k}}{\partial\psi^{n+1,k}}$ using $\psi^{n+1,k}$.

2.  Computation of the system matrix $\mathbf{R}'$ and the residual $\mathbf{R}$.

3.  Computation of the system solution $\psi^{n+1,k+1}$.

4.  Check convergence. If convergence is achieved, exit.

*enddo*

**Next time step**
where *k* is the iteration counter and *maxit* the maximum number of iterations.

The time-adaptive algorithm consists of keeping the pressure head constant and changing the
time step length. The algorithm is described by the following:

**For a given time step n**

-    Computation of the variable $\theta^n$, the parameter $\mathbf{K}^n$ and their derivatives $\dfrac{d\theta^n}{d\psi^n}, \dfrac{\partial\mathbf{K}^n}{\partial\psi^n}$

using $\psi^n$.

*do k=1, maxit*

1.  Define a time step $\Delta t^{n+1,k}$ depending on the time stepping strategy.

2.  Computation of the system matrix $\mathbf{R}'$ and the residual $\mathbf{R}$.

3.  Computation of the system solution $\psi^{n+1,k+1}$.

4.  Check convergence. If convergence is achieved, exit.

*enddo*

**Next time step**

The main advantage of the alternative algorithm is its avoidance of the computation of the
variable $\theta$, the parameter $\mathbf{K}$ and their derivatives $\dfrac{d\theta}{d\psi}$ and $\dfrac{\partial\mathbf{K}}{\partial\psi}$ during the iterations. Due to





the highly nonlinear relations between $\boldsymbol{\theta}$, $\mathbf{K}$, $\dfrac{d\boldsymbol{\theta}}{d\boldsymbol{\psi}}$, $\dfrac{\partial \mathbf{K}}{\partial \boldsymbol{\psi}}$ and the pressure, this computation
may require significant CPU time.

The most popular time step management during the simulation is that of the heuristic type
(Miller et al., 2006). The time step $\Delta t^{n+1}$ is computed depending on $\Delta t^{n}$ and the number of
iterations $k$ necessary to reach convergence in the following way:

$$
\begin{cases}
if \ \ k \leq m_1 & \Delta t^{n+1} = k_1 \Delta t^n & k_1 > 1.0 \\
if \ \ m_1 \leq k \leq m_2 & \Delta t^{n+1} = \Delta t^n & \\
if \ \ m_2 \leq k & \Delta t^{n+1} = k_2 \Delta t^n & k_2 < 1.0
\end{cases}
\tag{19}
$$

where $k_1$, $k_2$, $m_1$, $m_2$ are user-defined constants.
Other heuristic time step management procedures have been suggested by Kirkland et al.,
(1992) based on the water volumes exchanged between the adjacent cells of the grid and by
Ross (2003), where the time step size is controlled by the maximum allowed change in the
saturation.
For the Ross method, the fluxes are computed first and the time step magnitude is calculated
accordingly using

$$
\Delta t^{n+1} = \frac{\Delta S_{max}}{\max_i \left( \dfrac{\left| q_{-,i}^n - q_{+,i}^n \right|}{\Delta z_i \left( \theta_{s,i} - \theta_{r,i} \right)} \right)}
\tag{20}
$$

where $\Delta S_{max}$ is the user-defined maximum saturation change. After the computation of the
actual change in the saturation, the time step is modified if the maximum of the actual change
exceeds $(1+\lambda)\, max_i \left( \left| \Delta S_i \right| \right)$ where $\lambda$ is a user-defined value, according to:

$$
\Delta t^{n+1,k} = \frac{\Delta S_{max}}{\max_i \left( \left| \Delta S_i \right| \right)} \Delta t^{n+1,k-1}
\tag{21}
$$

and the system of equations is solved again. More details about handling the fluxes at
boundaries and saturated conditions can be found in Crevoisier et al. (2009), Ross (2003) and
Varado et al. (2006b).

Adaptive time stepping strategies based on time truncation error control were found to be
superior to others approaches (Hirthe and Graf, 2012; Kavetski et al., 2001; Tocci et al.,
1997). The Method of Lines using the DASPK integrator was applied to the Richards'
equation by Matthews et al. (2004), Miller et al. (1998), Tocci et al. (1997) among others. The
Method of Lines consists of discretization of the spatial part of the PDE only, leading to a
system of ordinary differential equations. It has been found to be significantly more efficient
than other temporal discretizations (Miller et al., 2006). However, Kavetski and Binning
(2002b) reported difficulties in obtaining convergence for the DASPK solver associated with
an arithmetic mean of inter-block conductivities for the most difficult problem addressed by
Miller et al. (1998).
The adaptive scheme used in this work evaluates the time steps through truncation error due
to the temporal discretization as proposed by Thomas and Gladwell (1988). This scheme was
already applied to the pressure-based formulation by Kavetski et al. (2001) and to the
moisture-based formulation by Kavetski and Binning (2004).
The difference between the first-order and second-order time approximations can be
considered as an estimate of the local truncation error of the first-order scheme. The first-
order approximation is given by:

$$\boldsymbol{\psi}_{(1)}^{n+1} = \boldsymbol{\psi}^{n} + \Delta t^{n+1} \frac{\partial \boldsymbol{\psi}^{n}}{\partial t} \tag{22}$$

The second-order approximation is:

$$\begin{aligned} \boldsymbol{\psi}_{(2)}^{n+1} &= \boldsymbol{\psi}^{n} + \Delta t^{n+1} \frac{\partial \boldsymbol{\psi}^{n}}{\partial t} + \frac{1}{2}\left(\Delta t^{n+1}\right)^{2} \frac{\partial^{2} \boldsymbol{\psi}^{n}}{\partial t^{2}} \\ &= \boldsymbol{\psi}^{n} + \frac{1}{2}\left(\Delta t^{n+1}\right)\left[ \frac{\partial \boldsymbol{\psi}^{n+1}}{\partial t} + \frac{\partial \boldsymbol{\psi}^{n}}{\partial t} \right] \end{aligned} \tag{23}$$

using $\dfrac{\partial \boldsymbol{\psi}^{n+1}}{\partial t} = \dfrac{\partial \boldsymbol{\psi}^{n}}{\partial t} + \Delta t^{n+1} \dfrac{\partial^{2} \boldsymbol{\psi}^{n}}{\partial t^{2}}$ .
This truncation error is given by:



$$\varepsilon_t^{n+1} = \max_i \left| \psi_{(2),i}^{n+1} - \psi_{(1),i}^{n+1} \right| = \frac{1}{2} \Delta t^{n+1} \max_i \left| \frac{\partial \psi_i^{n+1}}{\partial t} - \frac{\partial \psi_i^{n}}{\partial t} \right|$$

$$\approx \frac{1}{2} \Delta t^{n+1} \max_i \left| \frac{\psi_i^{n+1} - \psi_i^{n}}{\Delta t^{n+1}} - \frac{\psi_i^{n} - \psi_i^{n-1}}{\Delta t^{n}} \right| \qquad (24)$$


When the truncation error is smaller than γ, the temporal truncation error tolerance defined by
the user, the size of the next time step is calculated by:

$$\Delta t^{n+1} = \Delta t^n \min\left( s \sqrt{\frac{\gamma}{\max(\varepsilon_t^{n+1}, EPS)}}, r_{max} \right) \qquad (25)$$


When the truncation error is superior to γ, the computation is repeated with a reduced time
step defined as following:

$$\Delta t^n = \Delta t^n \max\left( s \sqrt{\frac{\gamma}{\max(\varepsilon_t^{n+1}, EPS)}}, r_{min} \right) \qquad (26)$$


where $r_{max}$ and $r_{min}$ are user-defined constants used to avoid too drastic changes of the time
step. $s$ is considered to be a safety factor that ensures that the time step changes are
reasonable. *EPS* is used to avoid floating point errors when the truncation error becomes too
small.

**5. Evaluation of the algorithms' performance**
We applied the NR method to the mixed form of RE using the standard iterative algorithm
and the time-adaptive algorithm. Implicit standard finite volumes have been used to solve the
partial differential equation and arithmetic means are used to compute the inter-block
hydraulic conductivity. The detailed discretizations of the matrix $\mathbf{R}'(\psi^{n+1,k})$ and the vector
$\mathbf{R}(\psi^{n+1,k})$ (see Eq. (18)) are given in Appendix 1. The time-adaptive algorithms have been





applied as described by the authors: Ross (2003) for the time stepping based on the saturation
changes and Kavetski et al. (2001) for the time stepping based on the truncation errors.
For the standard iterative algorithm, we defined two types of errors to check the convergence:
the error based on the maximum change of the state variables between two iterations defined
by $\varepsilon_\psi = \max_i \left| \psi_i^{n+1,k+1} - \psi_i^{n+1,k} \right|$ and the truncation error $\varepsilon_t$ defined by Eq. (24). Convergence is
assumed to be achieved when:
$$\varepsilon_\psi < \tau_a + \tau_r \left| \psi_{imax}^{n+1,k+1} \right| \qquad (27)$$
where $\tau_a$ and $\tau_r$ are the absolute and relative user-defined tolerances and $\psi_{imax}^{n+1,k+1}$ is the
pressure corresponding to $\varepsilon_\psi$ and when:
$$\varepsilon_t < \tau_a + \tau_r \left| \psi_{imax}^{n+1,k+1} \right| \qquad (28)$$
where the parameters have the same meaning as those for the previous criterion but $\psi_{imax}^{n+1,k+1}$
represents the pressure value corresponding to $\varepsilon_t$.
The tested algorithms are summarized in Table 1. Computations of all possible combinations
for the standard iterative scheme have been performed. We present only the four most
efficient algorithms.
We investigated three one-dimensional problems with various initial and boundary conditions
and hydraulic functions to assess the accuracy, efficiency and computational costs of the
different algorithms. The selected test cases represent a range of difficult infiltration problems
widely analyzed in the literature:
-   TC1: infiltration in a homogeneous initially dry soil with constant prescribed pressure

at the surface and prescribed pressure at the bottom (Celia et al., 1990);





278  -  TC2: infiltration in a homogeneous soil initially at hydrostatic equilibrium with a

279    prescribed constant flux at the soil surface and prescribed pressure at the bottom

280    (Miller et al., 1998);

281  -  TC3: infiltration/evaporation in an initially dry heterogeneous soil, with variable

282    positive and negative fluxes at the surface and free drainage at the base of the soil

283    column (Lehmann and Ackerer, 1998).

284 For the three test cases, the soil hydraulic functions were described by Mualem-Van

285 Genuchten models (Mualem, 1976; van Genuchten, 1980), see Eq. (4) and (5).

286 The required parameters, boundary conditions and initial conditions are summarized in Table

287 2. The evolution of the relative hydraulic conductivity, the water saturation and the specific

288 moisture capacity with respect to the pressure values are shown in Figures 1, 2 and 3,

289 respectively. For TC1, the pressure will vary from -1000 cm to -75 cm only due to the

290 specific conditions of this test case. Therefore, the parameter variations are smaller than those

291 for the other test cases. Since the parameters' variations are more abrupt for test cases 2 and 3,

292 their solutions are more challenging.

293 Preliminary tests were performed to define the optimal spatial discretization. We assume that

294 the errors are only originated from the time step size and the linearization.

295 The following criteria were used for the time stepping strategy:

296  -  $k_1$=0.80, $k_2$=1.20, $m_1$=5, $m_2$=10, which are the usual values for the heuristic strategy

297    defined by Eq. (19);

298  -  $r_{min}$=0.10, $r_{max}$=4.0, s=0.9, $EPS$=$10^{-10}$, which are the standard values for the time

299    stepping scheme based on time discretization error defined by Eq. (26) (Kavetski et

300    al., 2001);





- the maximum change in saturation has been evaluated using the maximum change in
the pressure according to the following relationship:
$$\Delta S_{max} \approx \frac{1}{\left( \theta_{s,imax} - \theta_{r,imax} \right)} \left. \frac{d\theta}{d\psi} \right|_{imax}^{n} \left( \tau_a + \tau_r \left| \psi_{imax}^{n+1,k+1} \right| \right) \qquad (29)$$

The simulations have been performed using different values of $\tau_r$ and with $\tau_a = 0.0$.

We used several criteria to evaluate the performance of these codes. A typical error used in
solving RE is the global cumulative mass balance error defined by:
$$\text{MB}(t^{n+1}) = \frac{\sum_{i=1}^{M} \Delta z_i \left( \theta_i^{n+1} - \theta_i^{0} \right)}{\sum_{k=1}^{n+1} \left( q_{in}^{k} - q_{out}^{k} \right) \Delta t^{k}} \qquad (30)$$

where $\Delta z_i$ is the size of the cell/element $i$, $\theta_i^{n+1}$ is its water content at time $t^{n+1}$, $\theta_i^{0}$ is the
initial water content, and $q_{in}^{k}$ and $q_{out}^{k}$ are the inflow and outflow, respectively, at the domain
boundaries at time $t^{k}$. $M$ is the number of cells/elements. The fluxes at the boundaries are
defined by $q^{k} = \frac{1}{2} \left( q^{k} + q^{k-1} \right)$. The mass balance errors were checked for each runs but were
found to be negligible since we solved the mass-conserving RE form.
While it is necessary to satisfy the global mass balance for an accurate numerical scheme, a
low mass balance error is not sufficient to ensure the accuracy of the solution. Therefore,
solutions have also been compared with the reference solution obtained using a very fine
temporal discretization and the iterative Newton-Raphson method. This comparison is based
on the average relative error defined by:



$$\varepsilon_k = \left[ \frac{1}{M} \sum_i \frac{\left| \psi_i^{ref} - \hat{\psi}_i \right|^k}{\left| \psi_i^{ref} \right|^k} \right]^{1/k}$$
(31)

where $M$ is the number of cells, $\boldsymbol{\psi}^{ref}$ is the reference solution and $\hat{\boldsymbol{\psi}}$ is the tested numerical
solution. $\varepsilon_1$ represents the average absolute relative error (called $L_1$-norm in the following),
$\varepsilon_2$ is the average quadratic error ($L_2$-norm) and $\varepsilon_\infty$ is the highest local relative difference
between the numerical and the reference solutions ($L_\infty$-norm).
Since the time-adaptive algorithm does not require the computation of the parameters and
their derivatives during the iterative procedure, we use $N_{sol}$ to denote the number of times
where the system of equations is solved and $N_{param}$ to denote the number of times where the
parameters are computed. Of course, these counters are equal to each other for the standard
algorithm and $N_{param}$ is less than $N_{sol}$ for the time-adaptive algorithm. For comparison
purposes, the computational costs are estimated by $N_{sol}$ for the standard algorithm and by ($N_{sol}$
$+N_{param}$)/2 for the time-adaptive algorithm. The efficiency of the algorithms have been
evaluated by comparing the computational costs for a given relative tolerance $\tau_r$. The errors
are presented in the tables and the figures. The figures show some additional results not listed
in the tables that already contains much information.

*TC1: Infiltration in a homogenous soil with constant boundary conditions*
This test case simulates an infiltration into a homogeneous porous medium. This problem is
addressed here because it has been widely analyzed previously by many authors like
Bouchemella et al. (2015), Celia et al. (1990), El Kadi and Ling (1993), Rathfelder and
Abriola, (1994), Tocci et al. (1997), among others. The computations were performed with a





spatial discretization of 0.1 cm. The initial time step size was set to $1.0 \ 10^{-5}$ s, and the
maximum time step size was set to 400 s.
The results for the iterative and time-adaptive algorithms are presented in Tables 3 and 4,
respectively. When both convergence criteria are used (algorithms SH_$\Delta\psi$_$\Delta$t and
SS_$\Delta\psi$_$\Delta$t), $N_{trunc}$ represents the number of times where the truncation error is the most
restrictive condition. For the heuristic time stepping schemes, the convergence is mostly
linked to the truncation error ($N_{trunc}$ is close to $N_{sol}$), whereas when the saturation time
stepping scheme is used, the most restrictive criterion is the maximum difference in the
pressure.
When the time stepping scheme is based on saturation, for both iterative and time-adaptive
algorithms, the number of iterations required to solve the problem is proportional to the
relative tolerance. Therefore, highly accurate solutions incur high computational costs.
For the time-adaptive scheme, the number of parameter changes $N_{param}$ is close to the number
of iterations for low tolerance values. Small tolerance values lead to small time steps,
avoiding time step adjustments. This is not the case for larger tolerance values that lead to
larger time steps and therefore to additional iterations (see for example TA_T for the
tolerance of $\tau_r = 10^{-2}$ – Table 4).
The three types of errors provide the same information. The best solution for one type of error
is also the best solution for the two others.
On average, the iterative algorithm is faster than the time-adaptive algorithm that requires
more iterations for a given error. This is also shown in Figure 4 that presents the convergence
rate of the $L_2$-norm with respect to the computational costs, *i.e.,* the number of iterations or
number of iterations and number of parameter changes. The time-adaptive algorithm with
time stepping based on the truncation errors performs quite poorly compared to the other





algorithms. Irrespective of the tolerance, this algorithm leads to a wetting front moving faster
(Fig. 5).
When the relative tolerance is set to a very low value ($\tau_r = 10^{-5}$), the iterative scheme with
time stepping based on the saturation changes shows behavior that is different from that found
for the less restrictive tolerance. The criterion based on truncation errors is no longer
significant ($N_{trunc}=252$), possibly explaining why the accuracy of the scheme remains
constant. This also indicates that errors due to time discretization have to be handled, either in
the convergence criterion or in the time stepping strategy.
For this test case, the most efficient algorithms are the iterative algorithms using the time
stepping strategy based on truncation error (ST_Δψ) or based on the saturation changes
(SS_Δψ_Δt ), except for the case of very high precision where ST_Δψ outperforms the other
algorithms.

*TC2: Infiltration in a homogenous soil with hydrostatic initial conditions*
This test case models an infiltration in a 200 cm vertical column of unconsolidated clay loam
with non-uniform grain size distribution and was considered by Miller et al. (1998) to be a
very challenging test. This problem was found to be more challenging from the numerical
point of view compared to TC1 due to the relative permeability function that enhances the
non-linear behavior of Richards' equation (Fig. 1, 2, 3). The cell size has been set to 0.125
cm, the initial time step to $10^{-5}$s and the maximum time step magnitude to 1000 s.
The different norms for the iterative and the time-adaptive schemes are given in Tables 5 and

6.

Investigation of this test case leads to similar qualitative conclusions when the time stepping
scheme is based on the saturation differences (SS_Δψ_Δt and TA_S). The standard scheme





SH_Δψ fails to provide an accurate solution within a reasonable number of iterations (less
than $10^7$).
The most efficient methods are the schemes using the time stepping strategy based on
truncation errors (Fig. 6). However, as found for TC1, the adaptive time algorithm failed to
provide highly accurate results ($L_2$-norm error less than approximately 4.5 $10^{-4}$).
Figure 7 shows the time step magnitudes for approximately equal $L_2$-norms for the two time-
adaptive algorithms and for the iterative algorithm using truncation errors for time stepping
(4.254 $10^{-4}$ within 3503 iterations for ST_Δψ, 4.563 $10^{-4}$ within 3094 iterations for TA_T and
4.844 $10^{-4}$ within 113583 iterations for TA_S). The increase in the time step length after 10 s
is the same, irrespective of the algorithm. For a smaller time, both truncation time stepping
strategies differ for the estimate of the first time step only. The scheme using the saturation
based time stepping is penalized by the poor estimate of the first maximum allowed saturation
change. This leads to the estimate of the first time step magnitude that was too long for
reaching convergence.

*TC3: Infiltration/evaporation in a heterogeneous soil*
This case study simulates infiltration in an initially dry heterogeneous soil with a succession
of rainfall and evaporations as upper boundary conditions during 35 days. This problem
differs from the two previous cases by the soil heterogeneity and also by the non-monotonic
boundary conditions at the soil surface. It is expected that non-monotonic discontinuous
boundary conditions will increase the difficulty of finding accurate solutions. The soil profile
consists of three 60 cm thick layers. The layers are discretized using cells with the size of 0.10
cm. The maximum time step magnitude is chosen as 0.20 days to avoid a too rough
discretization of the upper boundary conditions. The initial time step is set to $10^{-5}$ day.





The relative errors estimated by the iterative algorithms and the time-adaptive algorithms are
presented in Tables 7 and 8, respectively, and are plotted in Figure 8.
The standard iterative scheme fails to converge within the maximum number of iterations
($10^7$) when the tolerance is not sufficiently restrictive. The detailed analyses of the
computation showed that the time step size was quite large compared to the more restrictive
conditions until day 28.0 where the infiltration fluxes were equal to 1.50 cm/day and where
the conditions were near saturation due to the previous infiltration period. This led to a
decrease of the time step to close to the minimum value ($10^{-8}$ s), causing the procedure to
stop. More restrictive conditions lead to smaller time steps from the beginning of the
simulation and a better approximation of the solutions during the entire simulation.
The iterative scheme coupled with the truncation based time step strategy showed a
surprisingly unstable behavior for $\tau_r = 10^{-3}$. The scheme did not converge for
$\tau_r \in \left[ 0.96\,10^{-3}; 1.04\,10^{-3} \right]$. The results presented in Table 7 and Figure 8 are obtained for
$\tau_r = 0.90\,10^{-3}$. At this stage of our work, we were not able to provide a meaningful
explanation for this effect.
The time-adaptive algorithm with the saturation based time stepping scheme is the most
efficient for an $L_2$-norm greater than $10^{-4}$. For more accurate results, the iterative method with
the time stepping strategy using the truncation error must be preferred. The impact of the time
stepping strategy for these two algorithms is shown in Figure 9 for approximately the same
$L_2$-norm (2.051 $10^{-3}$ within 1283 iterations for TA_S and 1.517 $10^{-3}$ within 6504 iterations for
ST_Δψ). The time step changes is related to the boundary conditions variations as expected.
The strategy based on the saturation variation leads to a longer time step than the strategy
using the time truncation error. This difference can be quite important (see the simulation



between days 25 and 30). The consequences of this difference are a reduced number of
iterations but also a less accurate computation, irrespective of the error norm.

**6. Summary and conclusions**

The solution of RE is complex and very time consuming due to its highly non-linear
properties. Several algorithms have been tested for the mixed-form of Richards equation,
including time-adaptive methods. Based on the numerical examples that differ in their
parameters (level of non-linearity) and in their initial and boundary conditions, the
conclusions and recommendations are:
1.  Our numerical developments showed that the method suggested by Ross (2003) in its

implicit formulation can be considered as a Newton-Raphson method with a time-

adaptive algorithm.

2.  The different algorithms have different convergence rates (accuracy improvement of

the scheme as a function of the computational costs). Therefore, an algorithm can be

very efficient for a given accuracy and less efficient for another level of precision.

3.  The mass balance is not a good criterion for the evaluation of the results because the

mixed-form preserves the mass balance, irrespective of the pressure distribution

within the profile.

4.  The use of both criteria ($\varepsilon_\psi$, the maximum variable difference between two iterations,

$\varepsilon_t$ the time truncation error) should be implemented in the iterative procedure. The use

of $\varepsilon_\psi$ only, which is the case in many numerical codes, does not provide any

information about the accuracy of the time derivative approximation.

5.  Our 1-dimensional examples did not show a significant advantage of the time-adaptive

algorithm that avoids the computation of the parameters for each iteration. However,



this may depend on the number of elements used for the spatial discretization, and this
conclusion may be different for 2D or 3D domains.

Depending on the type of the problem that must be solved (parameters behavior with respect
to the pressure, time variations of the boundary conditions), the time truncation errors may be
predominant compared to the error corresponding to the pressure changes between two
iterations. Therefore, we recommend the use of both types of errors by implementing the
truncation errors either in the convergence procedure (convergence reached if $\varepsilon_\psi$ and $\varepsilon_t$ are
smaller than a user's defined tolerance) or in the time stepping strategy as defined by
Kavetski et al. (2001).




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



**APPENDIX 1.**
The numerical method used in the paper is implicit standard finite difference. For a cell *i* of
the grid, the unsaturated flow equation (4) can be discretized in the following way:

$$
\begin{cases}
\dfrac{\theta_i^{n+1} - \theta_i^n}{\Delta t} + S_w s_0 \dfrac{\psi_i^{n+1} - \psi_i^n}{\Delta t} + \dfrac{q_{i+}^{n+1} - q_{i-}^{n+1}}{\Delta z_i} = f_i \\[3mm]
q_{i-}^{n+1} = -K_{i-}\left( \dfrac{\psi_i^{n+1} - \psi_{i-1}^{n+1}}{\Delta z_{i-}} - 1 \right) \\[3mm]
q_{i+}^{n+1} = -K_{i+}\left( \dfrac{\psi_{i+1}^{n+1} - \psi_i^{n+1}}{\Delta z_{i+}} - 1 \right)
\end{cases}
\tag{A32}
$$

where *n* is the time step, $K_{i-}$ is the inter-block conductivity between cell *i* and (*i-1*) defined by
$K_{i-} = \dfrac{\Delta z_{i-1} K(\psi_{i-1}) + \Delta z_i K(\psi_i)}{\Delta z_{i-1} + \Delta z_i}$, $K_{i+}$ is the inter-block conductivity between cell *i* and (*i+1*)
defined by $K_{i+} = \dfrac{\Delta z_i K(\psi_i) + \Delta z_{i+1} K(\psi_{i+1})}{\Delta z_i + \Delta z_{i+1}}$. $\Delta z_{i-} = \dfrac{1}{2}\left(\Delta z_{i-1} + \Delta z_i\right)$ is the distance between the
center of cell (*i-1*) and *i*. $\Delta z_{i+} = \dfrac{1}{2}\left(\Delta z_i + \Delta z_{i+1}\right)$ is the distance between the center of cell *i* and
(*i+1*).
The residual is:
$R(\psi_i^{n+1,k}) = \Delta z_i\left(\theta_i^{n+1,k} - \theta_i^n\right) + \Delta z_i S_w s_0\left(\psi_i^{n+1,k} - \psi_i^n\right) + \Delta t\left(q_{i+}^{n+1,k} - q_{i-}^{n+1,k}\right) - \Delta t \Delta z_i f_i$ (A33)
where *k* is the iteration counter.
The residual derivatives are:

$$
\begin{aligned}
&\frac{\partial R(\psi_i^{n+1,k})}{\partial \psi_{i-1}^{n+1,k}} = -\Delta t\,\frac{\partial q_{i-}^{n+1,k}}{\partial \psi_{i-1}^{n+1,k}} \\[3mm]
&\frac{\partial R(\psi_i^{n+1,k})}{\partial \psi_i^{n+1,k}} = \Delta z_i\,\frac{d\theta_i^{n+1,k}}{d\psi_i^{n+1,k}} + \Delta z_i S_w s_0 + \Delta t\left(\frac{\partial q_{i+}^{n+1,k}}{\partial \psi_i^{n+1,k}} - \frac{\partial q_{i-}^{n+1,k}}{\partial \psi_i^{n+1,k}}\right) \\[3mm]
&\frac{\partial R(\psi_i^{n+1,k})}{\partial \psi_{i+1}^{n+1,k}} = \Delta t\,\frac{\partial q_{i+}^{n+1,k}}{\partial \psi_{i+1}^{n+1,k}}
\end{aligned}
\tag{A34}
$$

Therefore, the system to solve is:


$$-\Delta t \, \frac{\partial q_{i-}^{n+1,k}}{\partial \psi_{i-1}^{n+1,k}} \Delta \psi_{i-1}^{n+1,k+1} +$$

$$\left[ \Delta z_i \frac{d\theta_i^{n+1,k}}{d\psi_i^{n+1,k}} + \Delta z_i S_w s_0 + \Delta t \left( \frac{\partial q_{i+}^{n+1,k}}{\partial \psi_i^{n+1,k}} - \frac{\partial q_{i-}^{n+1,k}}{\partial \psi_i^{n+1,k}} \right) \right] \Delta \psi_i^{n+1,k+1} +$$

$$\Delta t \, \frac{\partial q_{i+}^{n+1}}{\partial \psi_{i+1}^{n+1,k}} \Delta \psi_{i+1}^{n+1,k+1} =$$

$$-\Delta z_i \left( \theta_i^{n+1,k} - \theta_i^n \right) - \Delta z_i S_w s_0 \left( \psi_i^{n+1,k} - \psi_i^n \right) - \Delta t \left( q_{i+}^{n+1,k} - q_{i-}^{n+1,k} \right) + \Delta t \Delta z_i f_i$$

(A35)

With the following derivatives of the fluxes $q_{i-}^{n+1,k}$

$$\begin{cases} \dfrac{\partial q_{i-}^{n+1,k}}{\partial \psi_{i-1}^{n+1,k}} = -\dfrac{\partial K_{i-}^{n+1,k}}{\partial \psi_{i-1}^{n+1,k}} \left( \dfrac{\psi_i^{n+1,k} - \psi_{i-1}^{n+1,k}}{\Delta z_{i-}} - 1 \right) + \dfrac{K_{i-}^{n+1,k}}{\Delta z_{i-}} \\[4mm] \dfrac{\partial q_{i-}^{n+1,k}}{\partial \psi_i^{n+1,k}} = -\dfrac{\partial K_{i-}^{n+1,k}}{\partial \psi_i^{n+1,k}} \left( \dfrac{\psi_i^{n+1,k} - \psi_{i-1}^{n+1,k}}{\Delta z_{i-}} - 1 \right) - \dfrac{K_{i-}^{n+1,k}}{\Delta z_{i-}} \end{cases}$$

(A36)

and $q_{i+}^{n+1,k}$ :

$$\begin{cases} \dfrac{\partial q_{i+}^{n+1,k}}{\partial \psi_i^{n+1,k}} = -\dfrac{\partial K_{i+}^{n+1,k}}{\partial \psi_i^{n+1,k}} \left( \dfrac{\psi_{i+1}^{n+1,k} - \psi_i^{n+1,k}}{\Delta z_{i+}} - 1 \right) + \dfrac{K_{i+}^{n+1,k}}{\Delta z_{i+}} \\[4mm] \dfrac{\partial q_{i+}^{n+1,k}}{\partial \psi_{i+1}^{n+1,k}} = -\dfrac{\partial K_{i+}^{n+1,k}}{\partial \psi_{i+1}^{n+1,k}} \left( \dfrac{\psi_{i+1}^{n+1,k} - \psi_i^{n+1,k}}{\Delta z_{i+}} - 1 \right) - \dfrac{K_{i+}^{n+1,k}}{\Delta z_{i+}} \end{cases}$$

(A37)

The component of the vector of the residuals **R** is given by equation (A33) and the coefficients of the matrix **R**' for cell $i$ are:

$$R'_{i-1,i} = \Delta t \left[ \frac{\partial K_{i-}^{n+1,k}}{\partial \psi_{i-1}^{n+1,k}} \left( \frac{\psi_i^{n+1,k} - \psi_{i-1}^{n+1,k}}{\Delta z_{i-}} - 1 \right) - \frac{K_{i-}^{n+1,k}}{\Delta z_{i-}} \right]$$

$$R'_{i,i} = \Delta z_i \left[ \frac{d\theta_i^{n+1,k}}{d\psi_i^{n+1,k}} + S_w s_0 \right] - \Delta t \left[ \frac{\partial K_{i+}^{n+1,k}}{\partial \psi_i^{n+1,k}} \left( \frac{\psi_{i+1}^{n+1,k} - \psi_i^{n+1,k}}{\Delta z_{i+}} - \right) - \frac{K_{i+}^{n+1,k}}{\Delta z_{i+}} \right]$$

(A38)

$$+\Delta t \left[ \frac{\partial K_{i-}^{n+1,k}}{\partial \psi_i^{n+1,k}} \left( \frac{\psi_i^{n+1,k} - \psi_{i-1}^{n+1,k}}{\Delta z_{i-}} - 1 \right) + \frac{K_{i-}^{n+1,k}}{\Delta z_{i-}} \right]$$

$$R'_{i,i+1} = -\Delta t \left[ \frac{\partial K_{i+}^{n+1,k}}{\partial \psi_{i+1}^{n+1,k}} \left( \frac{\psi_{i+1}^{n+1,k} - \psi_i^{n+1,k}}{\Delta z_{i+}} - 1 \right) + \frac{K_{i+}^{n+1,k}}{\Delta z_{i+}} \right]$$

In case of prescribed flux at the upper boundary, the residual is written as:



$$R_1(\psi_1^{n+1,k}) = \Delta z_1 \left[ \left( \theta_1^{n+1,k} - \theta_1^n \right) + S_w s_0 \left( \psi_1^{n+1,k} - \psi_1^n \right) \right] + \Delta t \left( q_{1+}^{n+1} - q_{BC} \right) - \Delta t \Delta z_1 f_1 \tag{A39}$$

Using the derivatives as defined in (A36) and (A37), the matrix coefficients are changed as
follow:

$$R'_{1,1} = \Delta z_1 \left( \frac{d\theta_1^{n+1,k}}{d\psi_1^{n+1,k}} + S_w s_0 \right) - \Delta t \left[ \frac{\partial K_{1+}^{n+1,k}}{\partial \psi_1^{n+1,k}} \left( \frac{\psi_2^{n+1,k} - \psi_1^{n+1,k}}{\Delta z_{1+}} - 1 \right) - \frac{K_{1+}^{n+1,k}}{\Delta z_{1+}} \right]$$
$$R'_{1,2} = -\Delta t \left[ \frac{\partial K_{1+}^{n+1,k}}{\partial \psi_2^{n+1,k}} \left( \frac{\psi_2^{n+1,k} - \psi_1^{n+1,k}}{\Delta z_{1+}} - 1 \right) + \frac{K_{1+}^{n+1,k}}{\Delta z_{1+}} \right] \tag{A40}$$


If the flux is applied at the bottom of the profile, similar developments lead to the residual:

$$R_N = \Delta z_N \left[ \left( \theta_N^{n+1,k} - \theta_N^n \right) + S_w s_0 \left( \psi_N^{n+1,k} - \psi_N^n \right) \right] + \Delta t \left( q_{BC} - q_{N-}^{n+1,k} \right) - \Delta t \Delta z_N f_N \tag{A41}$$


and its derivatives

$$R'_{N-1,N} = \Delta t \left[ \frac{\partial K_{N-}^{n+1,k}}{\partial \psi_{N-1}^{n+1,k}} \left( \frac{\psi_N^{n+1,k} - \psi_{N-1}^{n+1,k}}{\Delta z_{N-}} - 1 \right) - \frac{K_{N-}^{n+1,k}}{\Delta z_{N-}} \right]$$
$$R'_{N,N} = \Delta z_N \left( \frac{d\theta_N^{n+1,k}}{d\psi_N^{n+1,k}} + S_w s_0 \right) + \Delta t \left[ \frac{\partial K_{N-}^{n+1,k}}{\partial \psi_N^{n+1,k}} \left( \frac{\psi_N^{n+1,k} - \psi_{N-1}^{n+1,k}}{\Delta z_{N-}} - 1 \right) + \frac{K_{N-}^{n+1,k}}{\Delta z_{N-}} \right] \tag{A42}$$


If the pressure is described at the top of the soil, the corresponding flux is defined by:

$$q_{1-}^{n+1,k} = -K_{1-} \left( \frac{\psi_1^{n+1,k} - \psi_{BC}}{\Delta z_1 / 2} - 1 \right) \tag{A43}$$


And the derivative is:

$$\frac{\partial q_{1-}^{n+1,k}}{\partial \psi_1^{n+1,k}} = -\frac{\partial K_{1-}^{n+1,k}}{\partial \psi_1^{n+1,k}} \left( \frac{\psi_1^{n+1,k} - \psi_{BC}}{\Delta z_1 / 2} - 1 \right) - \frac{K_{1-}^{n+1,k}}{\Delta z_1 / 2} \tag{A44}$$


The corresponding residual and the matrix coefficients are:





$R_1 = \Delta z_1 \left[ \left( \theta_1^{n+1,k} - \theta_1^n \right) + S_w s_0 \left( \psi_1^{n+1,k} - \psi_1^n \right) \right] + \Delta t \left( q_{1+}^{n+1,k} - q_{1-}^{n+1,k} \right) - \Delta t \Delta z_1 f_1$     (A45)
and

$$R'_{1,1} = \Delta z_1 \left( \frac{d\theta_1^{n+1,k}}{d\psi_1^{n+1,k}} + S_w s_0 \right) - \Delta t \left[ \frac{\partial K_{1+}^{n+1,k}}{\partial \psi_1^{n+1,k}} \left( \frac{\psi_2^{n+1,k} - \psi_1^{n+1,k}}{\Delta z_{1+}} - 1 \right) - \frac{K_{1+}^{n+1,k}}{\Delta z_{1+}} \right]$$

$$+ \Delta t \left[ \frac{\partial K_{1-}^{n+1,k}}{\partial \psi_1^{n+1,k}} \left( \frac{\psi_1^{n+1,k} - \psi_{BC}}{\Delta z_1 / 2} - 1 \right) + \frac{K_{1-}^{n+1,k}}{\Delta z_1 / 2} \right]$$     (A46)

$$R'_{1,2} = -\Delta t \left[ \frac{\partial K_{1+}^{n+1,k}}{\partial \psi_2^{n+1,k}} \left( \frac{\psi_2^{n+1,k} - \psi_1^{n+1,k}}{\Delta z_{1+}} - 1 \right) + \frac{K_{1+}^{n+1,k}}{\Delta z_{1+}} \right]$$


Similarly, if the pressure is prescribed at the soils column's bottom, we have:

$R_N = \Delta z_N \left[ \left( \theta_N^{n+1,k} - \theta_N^n \right) + S_w s_0 \left( \psi_N^{n+1,k} - \psi_N^n \right) \right] + \Delta t \left( q_{N+}^{n+1,k} - q_{N-}^{n+1,k} \right) - \Delta t \Delta z_N f_N$     (A47)
and

$$R'_{N-1,N} = \Delta t \left[ \frac{\partial K_{N-}^{n+1,k}}{\partial \psi_{N-1}^{n+1,k}} \left( \frac{\psi_N^{n+1,k} - \psi_{N-1}^{n+1,k}}{\Delta z_{N-}} - 1 \right) - \frac{K_{N-}^{n+1,k}}{\Delta z_{N-}} \right]$$

$$R'_{N,N} = \Delta z_N \left( \frac{d\theta_N^{n+1,k}}{d\psi_N^{n+1,k}} + S_w s_0 \right) - \Delta t \left[ \frac{\partial K_{N+}^{n+1,k}}{\partial \psi_N^{n+1,k}} \left( \frac{\psi_{BC} - \psi_N^{n+1,k}}{\Delta z_N / 2} - 1 \right) - \Delta t \frac{K_{N+}^{n+1,k}}{\Delta z_N / 2} \right]$$     (A48)

$$+ \Delta t \left[ \frac{\partial K_{N-}^{n+1,k}}{\partial \psi_N^{n+1,k}} \left( \frac{\psi_N^{n+1,k} - \psi_{N-1}^{n+1,k}}{\Delta z_{N-}} - 1 \right) + \frac{K_{N-}^{n+1,k}}{\Delta z_{N-}} \right]$$


The numerical code is written in FORTRAN 90 and is available upon request.





**List of Tables**

Table 1. Different options of the tested algorithms. Reference to the corresponding equation in parenthesis.

Table 2: Domain size (L), initial conditions (IC), boundary conditions at the soil surface ($BC_u$) and at the soil bottom ($BC_l$), saturated hydraulic conductivity ($K_s$), residual and saturated water contents ($\theta_r$, $\theta_s$) and shape parameters ($\alpha$, $n$) for the different test cases. Length and time units are centimeters and seconds.

Table 3: Relative errors and number of iterations obtained for the iterative algorithm depending on different convergence criteria for TC1.

Table 4: Relative errors and number of iterations obtained for the time-adaptive algorithm depending on different convergence criteria for TC1.

Table 5: Relative errors and number of iterations obtained for the iterative algorithm depending on different convergence criteria for TC2 (n.c.: non convergence in less than $10^7$ iterations).

Table 6: Relative errors and number of iterations obtained for the time-adaptive algorithm depending on different convergence criteria for TC2.

Table 7: Relative errors and number of iterations obtained for the iterative algorithm depending on different convergence criteria for TC3 (n.c.: non convergence in less than $10^7$ iterations, * convergence failed for $10^{-3}$, $\tau_r$ =0.90 $10^{-3}$).

Table 8: Relative errors and number of iterations obtained for the time-adaptive algorithm depending on different convergence criteria for TC3.





| | Standard iterative algorithm | | | | | Time-adaptive algorithm | |
|---|---|---|---|---|---|---|---|
| | Time stepping | | | Stopping criterion | | | |
| | Heuristic (19) | Truncation (25) (26) | Saturation (20) (21) | Pressure (27) | Truncation (28) | Truncation (25) (26) | Saturation (20) (21) |
| SH_$\Delta\psi$ | x | | | x | | | |
| SH_$\Delta\psi$_$\Delta$t | x | | | x | x | | |
| ST_$\Delta\psi$ | | x | | x | | | |
| SS_$\Delta\psi$_$\Delta$t | | | x | x | x | | |
| TA_T | | | | | | x | |
| TA_S | | | | | | | x |

Table 1: Different options of the tested algorithms. Reference to the corresponding equation in parenthesis.

| | L | IC | $BC_u$ | $BC_l$ | $K_s$ | $\theta_r$ | $\theta_s$ | $\alpha$ | $\eta$ |
|---|---|---|---|---|---|---|---|---|---|
| TC1 | 30 | -1000.0 | $\psi=-75$ | $\psi=-1000$ | $9.22\ 10^{-3}$ | 0.102 | 0.368 | 0.0335 | 2.0 |
| TC2 | 200 | z-200 | $q=3.7\ 10^{-5}$ | $\psi=0$ | $7.18\ 10^{-5}$ | 0.095 | 0.410 | 0.019 | 1.31 |
| TC3 | 60 | -100.0 | q(t) | $q(t)=K_M(t)$ | $6.26\ 10^{-3}$ | 0.0286 | 0.366 | 0.028 | 2.239 |
| | 60 | -100.0 | | | $1.51\ 10^{-4}$ | 0.106 | 0.469 | 0.0104 | 1.395 |
| | 60 | -100.0 | | | $6.26\ 10^{-3}$ | 0.0286 | 0.366 | 0.028 | 2.239 |

Table 2: Domain size (L), initial conditions (IC), boundary conditions at the soil surface (BC$_u$) and at the soil bottom (BC$_l$), saturated hydraulic conductivity ($K_s$), residual and saturated water contents ($\theta_r$, $\theta_s$) and shape parameters ($\alpha$, $\eta$) for the different test cases.
$K_M(t)$ is the hydraulic conductivity of the last grid cell.
Length and time units are centimeters and seconds respectively.



| Tol. | Algorithm | $L_1$ | $L_2$ | $L_\infty$ | $N_{trunc}$ | $N_{sol}$ |
|---|---|---|---|---|---|---|
| $10^{-5}$ | SH_$\Delta\psi$ | $1.918\ 10^{-3}$ | $8.829\ 10^{-3}$ | $0.106$ | | $2177$ |
| | SH_$\Delta\psi$_$\Delta t$ | $8.391\ 10^{-6}$ | $6.459\ 10^{-5}$ | $8.782\ 10^{-4}$ | $542371$ | $615880$ |
| | ST_$\Delta\psi$ | $3.968\ 10^{-4}$ | $1.045\ 10^{-3}$ | $3.512\ 10^{-3}$ | | $6160$ |
| | SS_$\Delta\psi$_$\Delta t$ | $1.136\ 10^{-5}$ | $3.406\ 10^{-5}$ | $2.817\ 10^{-4}$ | $252$ | $3920446$ |
| $10^{-4}$ | SH_$\Delta\psi$ | $2.557\ 10^{-3}$ | $1.375\ 10^{-2}$ | $0.168$ | | $1701$ |
| | SH_$\Delta\psi$_$\Delta t$ | $7.818\ 10^{-5}$ | $2.259\ 10^{-4}$ | $1.593\ 10^{-3}$ | $170438$ | $194420$ |
| | ST_$\Delta\psi$ | $1.331\ 10^{-3}$ | $1.316\ 10^{-3}$ | $1.181\ 10^{-2}$ | | $1950$ |
| | SS_$\Delta\psi$_$\Delta t$ | $8.607\ 10^{-6}$ | $3.525\ 10^{-5}$ | $3.899\ 10^{-4}$ | $154597$ | $392041$ |
| $10^{-3}$ | SH_$\Delta\psi$ | $3.956\ 10^{-3}$ | $1.166\ 10^{-2}$ | $0.125$ | | $1312$ |
| | SH_$\Delta\psi$_$\Delta t$ | $2.320\ 10^{-4}$ | $7.553\ 10^{-4}$ | $7.883\ 10^{-3}$ | $52723$ | $60303$ |
| | ST_$\Delta\psi$ | $2.241\ 10^{-3}$ | $5.702\ 10^{-3}$ | $1.792\ 10^{-2}$ | | $620$ |
| | SS_$\Delta\psi$_$\Delta t$ | $6.567\ 10^{-5}$ | $1.585\ 10^{-4}$ | $1.453\ 10^{-3}$ | $9895$ | $39110$ |
| $10^{-2}$ | SH_$\Delta\psi$ | $6.559\ 10^{-3}$ | $1.716\ 10^{-2}$ | $0.119$ | | $1018$ |
| | SH_$\Delta\psi$_$\Delta t$ | $2.224\ 10^{-3}$ | $7.923\ 10^{-3}$ | $7.111\ 10^{-2}$ | $15540$ | $17888$ |
| | ST_$\Delta\psi$ | $9.954\ 10^{-3}$ | $2.630\ 10^{-2}$ | $8.727\ 10^{-2}$ | | $243$ |
| | SS_$\Delta\psi$_$\Delta t$ | $8.283\ 10^{-4}$ | $2.271\ 10^{-3}$ | $1.478\ 10^{-2}$ | $862$ | $3804$ |

Table 3: Relative errors and number of iterations obtained for the iterative algorithm depending on different convergence criteria for TC1.





| Tol. | Algorithm | $L_1$ | $L_2$ | $L_\infty$ | $N_{param}$ | $N_{sol}$ |
|---|---|---|---|---|---|---|
| $10^{-5}$ | **TA_T** | $5.016\ 10^{-3}$ | $2.376\ 10^{-2}$ | $0.269$ | $32197$ | $35938$ |
| | **TA_S** | $6.152\ 10^{-6}$ | $2.429\ 10^{-5}$ | $2.561\ 10^{-4}$ | $9316700$ | $9322946$ |
| $10^{-4}$ | **TA_T** | $5.598\ 10^{-3}$ | $2.580\ 10^{-2}$ | $0.284$ | $10169$ | $11520$ |
| | **TA_S** | $2.839\ 10^{-5}$ | $1.363\ 10^{-4}$ | $1.654\ 10^{-3}$ | $931616$ | $938144$ |
| $10^{-3}$ | **TA_T** | $1.524\ 10^{-2}$ | $7.085\ 10^{-2}$ | $0.822$ | $3231$ | $4032$ |
| | **TA_S** | $2.537\ 10^{-4}$ | $1.271\ 10^{-3}$ | $1.568\ 10^{-2}$ | $93114$ | $100898$ |
| $10^{-2}$ | **TA_T** | $6.241\ 10^{-2}$ | $0.274$ | $2.459$ | $1023$ | $1402$ |
| | **TA_S** | $2.519\ 10^{-3}$ | $1.224\ 10^{-2}$ | $0.142$ | $9267$ | $18292$ |

Table 4: Relative errors and number of iterations obtained for the time-adaptive algorithm depending on different convergence criteria for TC1.



| Tol. | Algorithm | $L_1$ | $L_2$ | $L_\infty$ | $N_{trunc}$ | $N_{sol}$ |
|---|---|---|---|---|---|---|
| $10^{-5}$ | **SH_$\Delta\psi$** | $6.966\ 10^{-3}$ | $1.818\ 10^{-2}$ | $5.878\ 10^{-2}$ | | 573 |
| | **SH_$\Delta\psi$_$\Delta t$** | $3.697.\ 10^{-4}$ | $9.766\ 10^{-4}$ | $3.332\ 10^{-3}$ | 53769 | 59643 |
| | **ST_$\Delta\psi$** | $1.578\ 10^{-4}$ | $4.254\ 10^{-4}$ | $2.451\ 10^{-3}$ | | 3503 |
| | **SS_$\Delta\psi$_$\Delta t$** | - | - | - | - | n. c. |
| $10^{-4}$ | **SH_$\Delta\psi$** | $6.966\ 10^{-3}$ | $1.818\ 10^{-2}$ | $5.878\ 10^{-2}$ | | 509 |
| | **SH_$\Delta\psi$_$\Delta t$** | $6.968\ 10^{-4}$ | $1.979\ 10^{-3}$ | $5.726\ 10^{-3}$ | 16557 | 18428 |
| | **ST_$\Delta\psi$** | $5.814\ 10^{-4}$ | $1.492\ 10^{-3}$ | $6.711\ 10^{-3}$ | | 1033 |
| | **SS_$\Delta\psi$_$\Delta t$** | $3.279\ 10^{-6}$ | $1.239\ 10^{-5}$ | $8.603\ 10^{-5}$ | 0 | 2474120 |
| $10^{-3}$ | **SH_$\Delta\psi$** | $6.966\ 10^{-3}$ | $1.818\ 10^{-2}$ | $5.878\ 10^{-2}$ | | 410 |
| | **SH_$\Delta\psi$_$\Delta t$** | $3.699\ 10^{-3}$ | $9.761\ 10^{-3}$ | $3.275\ 10^{-2}$ | 4830 | 5444 |
| | **ST_$\Delta\psi$** | $1.553\ 10^{-3}$ | $4.226\ 10^{-3}$ | $2.457\ 10^{-2}$ | | 317 |
| | **SS_$\Delta\psi$_$\Delta t$** | $2.355\ 10^{-5}$ | $6.230\ 10^{-5}$ | $2.341\ 10^{-4}$ | 0 | 247426 |
| $10^{-2}$ | **SH_$\Delta\psi$** | $6.892\ 10^{-3}$ | $1.800\ 10^{-2}$ | $5.780\ 10^{-2}$ | | 309 |
| | **SH_$\Delta\psi$_$\Delta t$** | $9.135\ 10^{-3}$ | $2.409\ 10^{-2}$ | $7.925\ 10^{-2}$ | 376 | 580 |
| | **ST_$\Delta\psi$** | $2.756\ 10^{-3}$ | $1.134\ 10^{-2}$ | $7.715\ 10^{-2}$ | | 180 |
| | **SS_$\Delta\psi$_$\Delta t$** | $2.973\ 10^{-4}$ | $7.884\ 10^{-4}$ | $3.252\ 10^{-3}$ | 0 | 24757 |

Table 5: Relative errors and number of iterations obtained for the iterative algorithm depending on different convergence criteria for TC2 (n.c.: non convergence in less than $10^7$ iterations).



| Tol. | Algorithm | $L_1$ | $L_2$ | $L_\infty$ | $N_{param}$ | $N_{sol}$ |
|---|---|---|---|---|---|---|
| $10^{-5}$ | **TA_T** | $1.230.\ 10^{-4}$ | $4.563\ 10^{-4}$ | $3.346\ 10^{-3}$ | 3089 | 3098 |
| | **TA_S** | $8.741\ 10^{-6}$ | $2.308\ 10^{-5}$ | $7.905\ 10^{-5}$ | 1136193 | 1136199 |
| $10^{-4}$ | **TA_T** | $1.572\ 10^{-3}$ | $4.497\ 10^{-3}$ | $2.404\ 10^{-2}$ | 986 | 987 |
| | **TA_S** | $2.701\ 10^{-5}$ | $7.219\ 10^{-5}$ | $3.095\ 10^{-4}$ | 113616 | 113616 |
| $10^{-3}$ | **TA_T** | $4.707\ 10^{-3}$ | $1.346\ 10^{-2}$ | $7.169\ 10^{-2}$ | 323 | 323 |
| | **TA_S** | $1.754\ 10^{-4}$ | $4.844\ 10^{-4}$ | $2.391\ 10^{-3}$ | 11358 | 11358 |
| $10^{-2}$ | **TA_T** | $5.220\ 10^{-3}$ | $1.683\ 10^{-2}$ | $0.101$ | 135 | 135 |
| | **TA_S** | $1.596\ 10^{-3}$ | $4.444\ 10^{-3}$ | $2.243\ 10^{-2}$ | 1132 | 1132 |

Table 6: Relative errors and number of iterations obtained for the time-adaptive algorithm depending on different convergence criteria for TC2.



| Tol. | Algorithm | $L_1$ | $L_2$ | $L_\infty$ | $N_{trunc}$ | $N_{sol}$ |
|---|---|---|---|---|---|---|
| | $SH\_\Delta\psi$ | $9.994\ 10^{-3}$ | $1.119\ 10^{-2}$ | $1.554\ 10^{-2}$ | | 1644 |
| $10^{-5}$ | $SH\_\Delta\psi\_\Delta t$ | $6.612\ 10^{-4}$ | $7.346\ 10^{-4}$ | $1.116\ 10^{-3}$ | 171636 | 190588 |
| | $ST\_\Delta\psi$ | $6.830\ 10^{-4}$ | $7.775\ 10^{-4}$ | $1.648\ 10^{-3}$ | | 16984 |
| | $SS\_\Delta\psi\_\Delta t$ | $7.185\ 10^{-5}$ | $7.935\ 10^{-5}$ | $1.297\ 10^{-4}$ | 197481 | 1646346 |
| | $SH\_\Delta\psi$ | $6.664\ 10^{-3}$ | $7.280\ 10^{-3}$ | $1.033\ 10^{-2}$ | | 1734 |
| $10^{-4}$ | $SH\_\Delta\psi\_\Delta t$ | $3.512\ 10^{-3}$ | $3.898\ 10^{-3}$ | $5.811\ 10^{-3}$ | 57312 | 63956 |
| | $ST\_\Delta\psi$ | $1.300\ 10^{-3}$ | $1.517\ 10^{-3}$ | $2.412\ 10^{-3}$ | | 6504 |
| | $SS\_\Delta\psi\_\Delta t$ | $5.380\ 10^{-5}$ | $6.536\ 10^{-5}$ | $1.010\ 10^{-4}$ | 41073 | 186351 |
| | $SH\_\Delta\psi$ | - | - | - | | n.c. |
| $10^{-3}$ | $SH\_\Delta\psi\_\Delta t$ | $2.625\ 10^{-3}$ | $2.899\ 10^{-3}$ | $4.971\ 10^{-3}$ | 22047 | 24779 |
| | $ST\_\Delta\psi$ | $4.730\ 10^{-3}$ | $5.422\ 10^{-3}$ | $1.036\ 10^{-2}$ | | 1297* |
| | $SS\_\Delta\psi\_\Delta t$ | $7.569\ 10^{-4}$ | $8.820\ 10^{-4}$ | $1.402\ 10^{-3}$ | 16474 | 31276 |
| | $SH\_\Delta\psi$ | - | - | - | | n.c. |
| $10^{-2}$ | $SH\_\Delta\psi\_\Delta t$ | $5.493\ 10^{-3}$ | $6.306\ 10^{-3}$ | $1.171\ 10^{-3}$ | 7438 | 8812 |
| | $ST\_\Delta\psi$ | $6.621\ 10^{-3}$ | $7.402\ 10^{-3}$ | $1.042\ 10^{-2}$ | | 810 |
| | $SS\_\Delta\psi\_\Delta t$ | $7.511\ 10^{-3}$ | $8.780\ 10^{-3}$ | $1.378\ 10^{-2}$ | 5838 | 7535 |

Table 7: Relative errors and number of iterations obtained for the iterative algorithm depending on different convergence criteria for TC3 (n.c.: non convergence in less than $10^7$ iterations, * convergence failed for $10^{-3}$, $\tau_r = 0.90\ 10^{-3}$).





| Tol. | Algorithm | $L_1$ | $L_2$ | $L_\infty$ | $N_{param}$ | $N_{sol}$ |
|---|---|---|---|---|---|---|
| $10^{-5}$ | **TA_T** | $9.814 \ 10^{-3}$ | $9.949 \ 10^{-3}$ | $1.286 \ 10^{-2}$ | 8369 | 8703 |
| | **TA_S** | $7.980 \ 10^{-5}$ | $8.797 \ 10^{-5}$ | $1.472 \ 10^{-4}$ | 1357075 | 1357160 |
| $10^{-4}$ | **TA_T** | $1.731 \ 10^{-2}$ | $1.760 \ 10^{-2}$ | $2.748 \ 10^{-2}$ | 2653 | 2934 |
| | **TA_S** | $1.067 \ 10^{-4}$ | $1.247 \ 10^{-4}$ | $1.997 \ 10^{-4}$ | 135386 | 135498 |
| $10^{-3}$ | **TA_T** | $2.922 \ 10^{-2}$ | $3.105 \ 10^{-2}$ | $4.545 \ 10^{-2}$ | 889 | 1153 |
| | **TA_S** | $1.433 \ 10^{-4}$ | $1.788 \ 10^{-4}$ | $3.367 \ 10^{-4}$ | 13314 | 13397 |
| $10^{-2}$ | **TA_T** | $1.996 \ 10^{-2}$ | $2.449 \ 10^{-2}$ | $5.536 \ 10^{-2}$ | 347 | 515 |
| | **TA_S** | $1.851 \ 10^{-3}$ | $2.051 \ 10^{-3}$ | $3.925 \ 10^{-3}$ | 1232 | 1283 |

Table 8: Relative errors and number of iterations obtained for the time-adaptive algorithm depending on different convergence criteria for TC3.





**List of Figures**

Figure 1: Relative hydraulic conductivity as a function of the pressure for the three test cases (L1, L2 and L3 are the three layers for test case 3).

Figure 2: Water saturation as a function of the pressure for the three test cases (L1, L2 and L3 are the three layers for test case 3).

Figure 3: Specific moisture capacity as a function of the pressure for the three test cases (L1, L2 and L3 are the three layers for test case 3).

Figure 4: Evolution of the $L_2$ relative error with computational costs for TC1.

Figure 5: Pressure profiles in the domain for the TA_T algorithm.

Figure 6: Evolution of the $L_2$ relative error with computational costs for TC2.

Figure 7: Time step magnitudes during the simulation for TC2.

Figure 8: Evolution of the $L_2$ relative error with computational costs for TC3.

Figure 9: Time step magnitudes during the simulation for TC3 for the time stepping strategy based on truncation error (TA_S in blue, TA_T in black, time varying boundary conditions at the top).

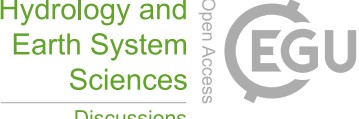

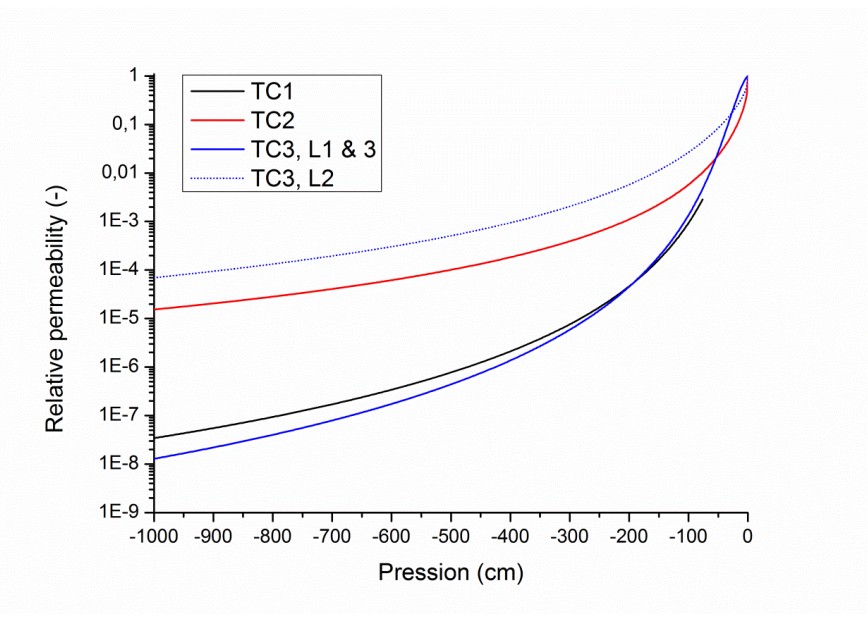

Figure 1: Relative permeability as a function of the pressure for the three test cases
(L1, L2 and L3 are the three layers for test case 3).





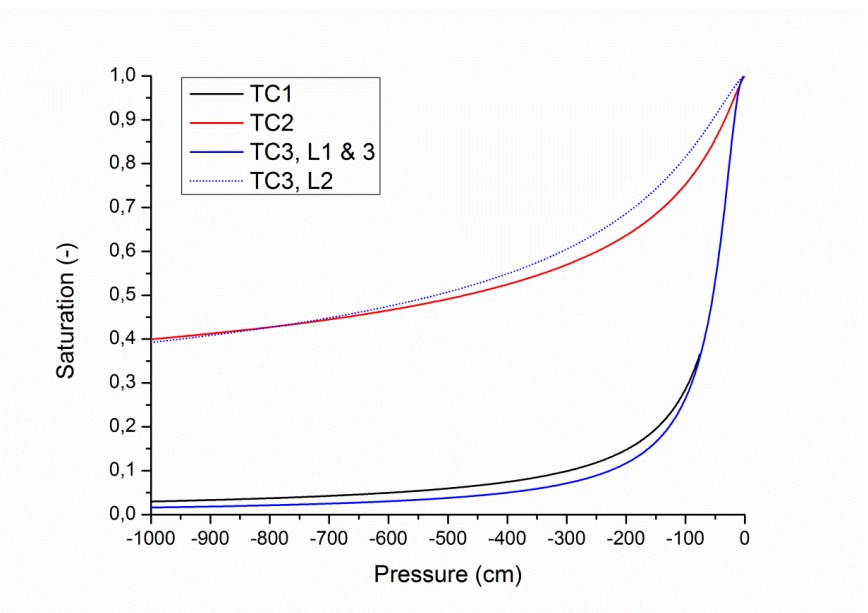

Figure 2: Water saturation as a function of the pressure for the three test cases
(L1, L2 and L3 are the three layers for test case 3).





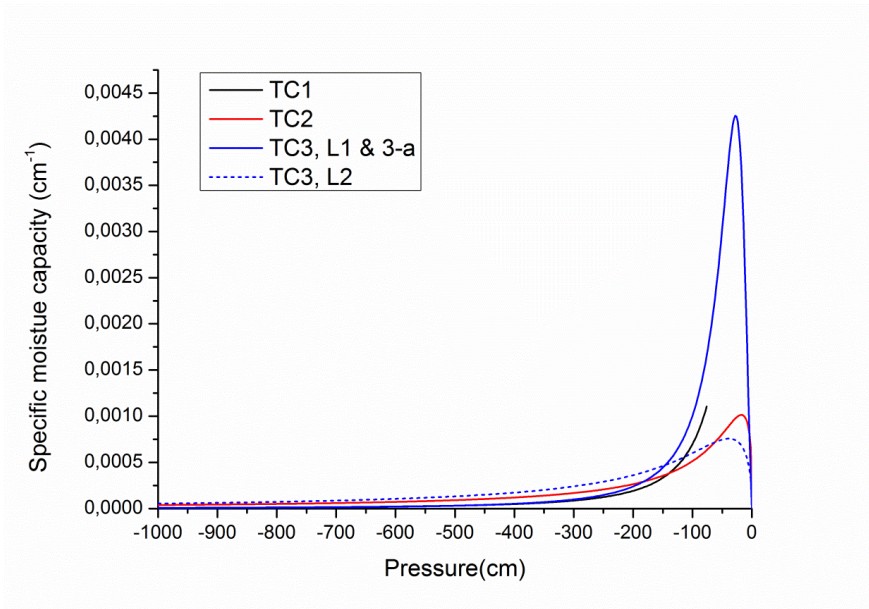

Figure 3: Specific moisture capacity as a function of the pressure for the three test cases
(L1, L2 and L3 are the three layers for test case 3).





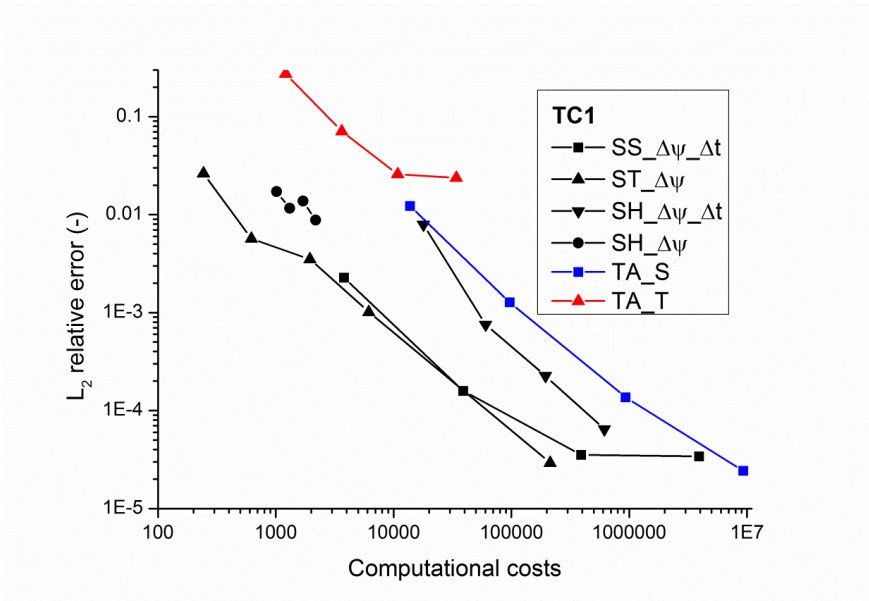

Figure 4: Evolution of the $L_2$ relative error with computational costs for TC1.





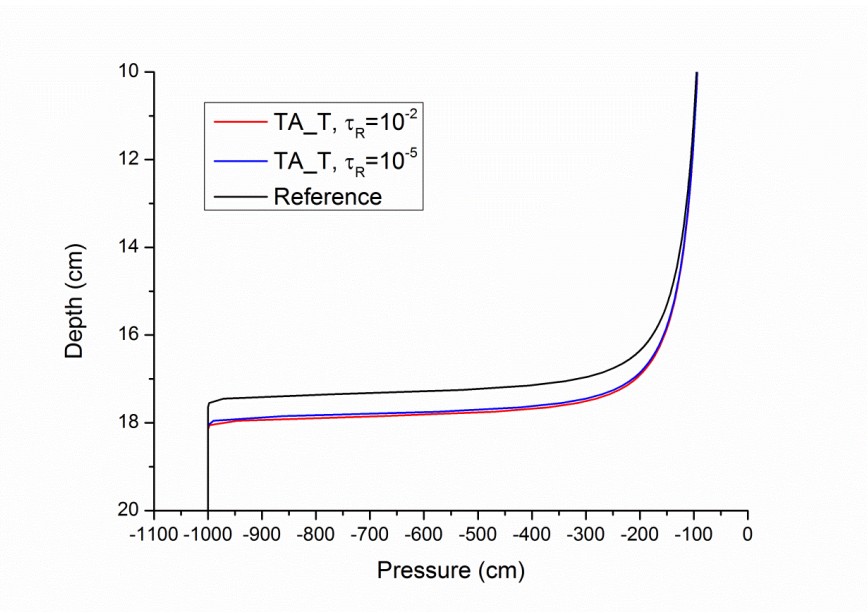

Figure 5: Pressure profiles in the domain for the TA_T algorithm.





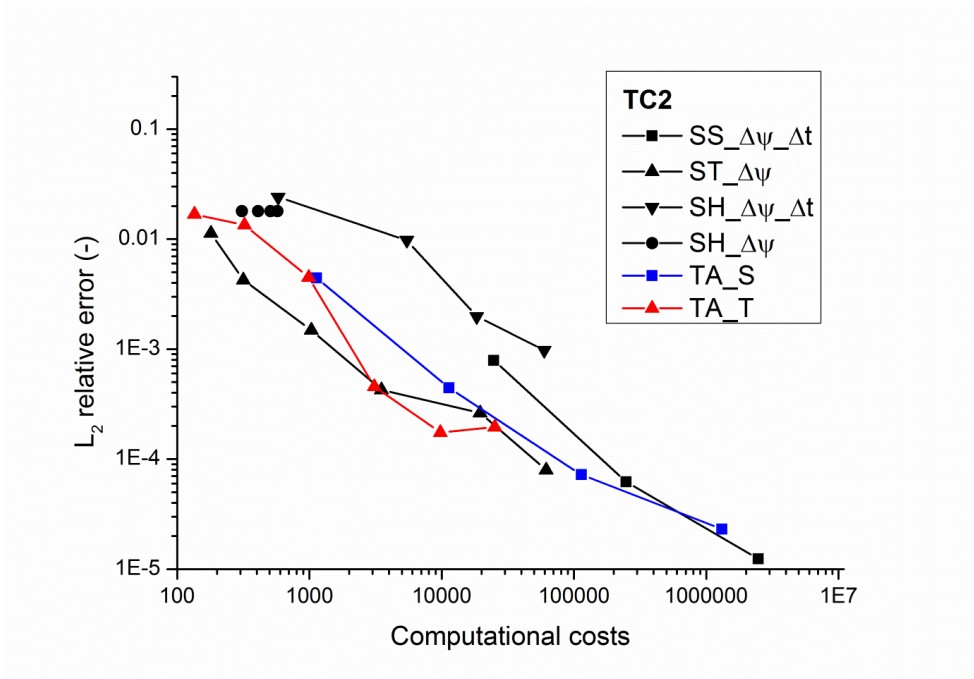

Figure 6: Evolution of the $L_2$ relative error with computational costs for TC2.





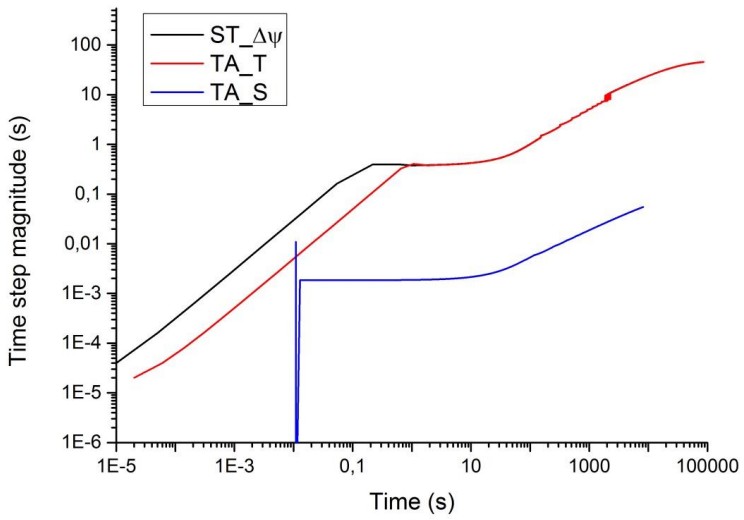

Figure 7: Time step magnitudes during the simulation for TC2.





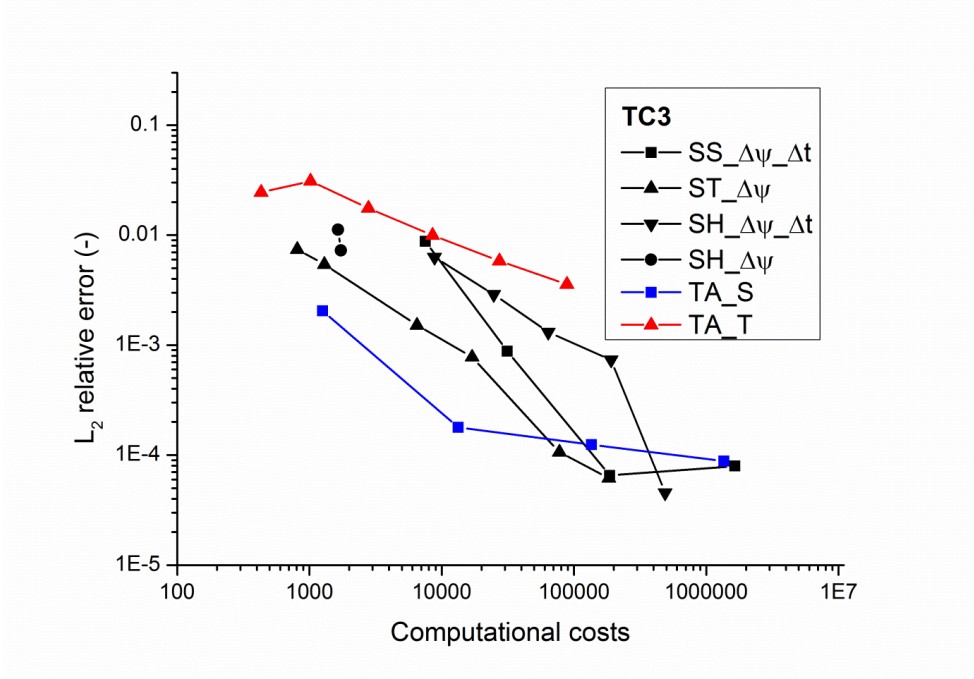

Figure 8: Evolution of the $L_2$ relative error with computational costs for TC3.





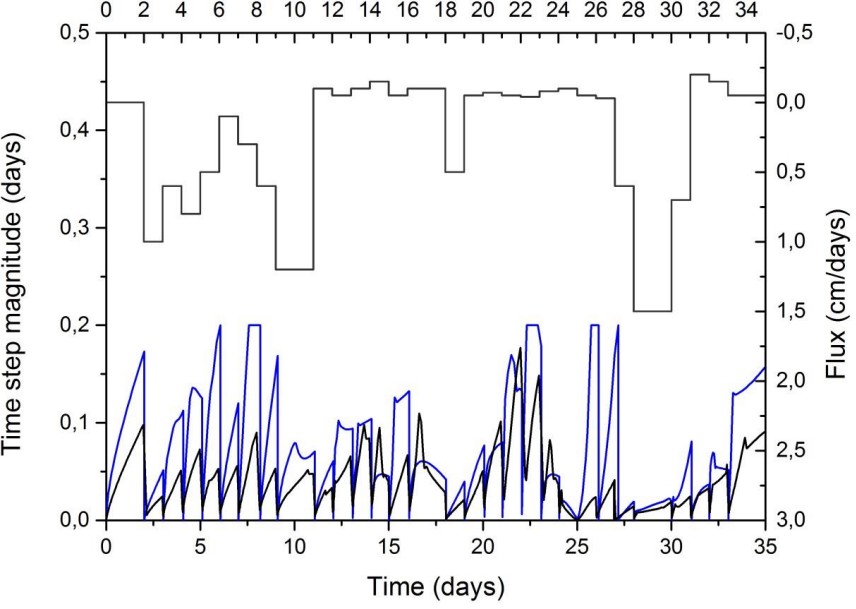

Figure 9: Time step magnitudes during the simulation for TC3 for the time stepping strategy based on truncation error (TA_S in blue, TA_T in black, time varying boundary conditions at the top).