# Peer review of "Ross scheme, Newton-Raphson iterative methods and time-stepping strategies for solving the mixed-form of Richards' equation"

_Hydrology and Earth System Sciences, 2016_

## Referee Comment (RC1) · Anonymous Referee #1 · 8 Mar 2017

This manuscript provides a performance assessment of different known algorithms to solve the Richard's equation. In particular, the authors investigate the performance of the Ross method versus the Newton-Raphson with different time-stepping strategies. A nice set of guidelines are provided in the end. The article is well written and provides a nice contribution in this area. Based on this, I suggest the publication of the manuscript after minor revision is addressed to tackle this points:

Minor comments:

- The author should clearly state the assumptions in Equation (1), rigid solid matrix (negligible changes in porosity) but also need to say that $\frac{1}{\rho}\nabla\rho \approx 0$. In this context, it is worth mention that the specific storage coefficient used in Equation (1) is not exactly the same as the specific storage coefficient of the flow equation. The specific storage coefficient is the sum of compressibility of water and soil. In equation (1) the changes in porosity are neglected and therefore "so" is not exactly the specific storage. Only the part corresponding to the compressibility of water.

- Line 36: actually there are three standard forms of the equation: pressure, saturation and mixed

- Equation (13), may be is worth to explain how to calculate fluxes q or simply refer to the appendix here for an example.

- Equation (15), maybe is worth explaining index k

- It is not clear whether the method suggested by Ross (2003) is mathematically equivalent to Newton-Raphson or simply performs the same way in this example. In case it is mathematically equivalent a more detail derivation is required. In case it performs equally in this case the manuscript should clearly state this fact. Could it be that in 2D and 3D the performance of these two algorithms is different?

---

## Referee Comment (RC2) · Anonymous Referee #2 · 14 Mar 2017

The paper compares the efficiency of a combination of two linearisation schemes for the solution of the non-linear Richards' equation with different time adaptation criteria. The first scheme is a method presented by Ross in 2003, a kind of semi-implicit scheme, calculating the non-linearities with the solution of the last time step. The second scheme is using Newton iterations. The authors first show, that if applied to the water content-formulation of Richards' equation, the method of Ross is equivalent to the first iteration of a Newton iteration. As the water-content form is only applicable to strictly unsaturated conditions, they use discretisations of the mixed form for the rest of the paper. In the Ross-type scheme, called time-adaptive (though both schemes use

adaptive time stepping), the authors apply only the first-iteration of a Newton-scheme, calculating the coefficients again with the old solution, and shorten the time step until convergence. In the Newton-iteration scheme they calculate the coefficients with the last iterate until convergence. Thus in the Ross-type scheme the assembly of the linear-equation system to be solved is faster for the second or later iterates. For the adaption of the time step the authors either use an heuristic approach based on the number of Newton iterations (only for the Newton-based scheme), an approach based on an estimation of the truncation error, or a limit on the maximal allowed change of saturation. The different combinations of time-step control and linearisation approach are applied to three different test cases from the literature. The computational costs, measured in a normalized number of solves, are plotted against precision, measured as the deviation of the results from a reference solution calculated with a very fine time step and a given grid size. The authors conclude, that there was no real advantage of the Ross-type scheme.

**General comments:**
The authors address a question, which has been intensively discussed in the last decades. Numerous papers on the best linearisation schemes and time-step adaptation procedures can be found easily in the literature, partially co-authored by one of the authors of this paper, many of them also cited in the paper. Thus the main question is, if the analysis of a very special scheme is a meaningful contribution to the literature and suited for publication in HESS. As there remain a lot of questions to be addressed (see specific comments below), the paper could be accepted only after major revisions. However, I am not convinced that the contributions made by the paper will be significant even after revision.

**Specific comments:**

- equation (1) + (2): As a rigid solid matrix is assumed, $s_0$ could only describe the

compressibility of the fluid. As water is nearly incompressible at the pressures occurring in variably saturated soils, the compressibility term is unnecessary and should be dropped.

- line 177: "The time-adaptive algorithm consists of keeping the pressure head constant and changing the time step length." Actually, this formulation is misleading. For each tested time step a new solution for the pressure heads is calculated. Thus they are not kept constant. However, the non-linear parameters are always calculated with the solution from the old time step, corresponding to a semi-implicit scheme. Even the matrix has to be reassembled for each tested time step. Thus only the evaluation of the non-linear functions is avoided. An alternative would be the use of an interpolation table for the hydraulic functions to reduce the computational costs and still keep the accuracy high. The misleading formulation is also used in line 6 of the abstract.

- line 210: I do not understand this formulation. $\max_i(|\Delta S_{\max}|)$ is the maximum of the actual change, how can it exceed itself? Do you mean exceeds $(1+\lambda)\Delta S_{\max}$?

- line 219-226: Is this important here? If necessary at all, please move it to the introduction

- line 243: replace "superior to" by "larger than"

- line 253: "Implicit standard finite volumes" is not really a precise description. I guess you mean a cell-centred finite volume scheme for the spatial discretisation with an implicit Euler-scheme for the temporal discretisation. Actually, already in chapter 3, equation (15) the discretisation is given. Shouldn't you just refer to that section?

- line 261: "the error based on the maximum change of the state variables between two iterations" would be $\max_{i,k}|\psi_i^{n+1,k+1} - \psi_i^{n+1,k}|$. If your formula is correct you

are looking at "the error base on the maximal change of the state variables in the last iteration". This actually is a very bad convergence condition as it can not distinguish between "already converged" and "no convergence at all". However, it is also completely unclear to me, why the time truncation error should be a sensible stopping condition. A reasonable stopping condition is based on the reduction of the non-linear residual compared to the initial non-linear residual. This would really be related to a reduction of the error in the solution of the non-linear equation.

- equation (27) and (28): is it really necessary to write out this equations? Is it not enough to state that relative and absolute error bounds are given?

- line 269-271: Actually, not all possible combinations have been performed. You could also have tested using only the truncation error (if this makes sense).

- line 293-294: as the spatial discretisation error (though not explicitly mentioned) is addressed here: How did you check, that the grid really was fine enough? As you try to get very accurate solutions in time (down to an error of $10^{-5}$), did you really make sure, that the grid is fine enough to produce changes significantly lower than $10^{-5}$ if further refined?

- line 301-303: As you are using a mixed scheme: why did you not just calculate $\Delta S_{\mathrm{max}}$ from the saturations? I am also a bit confused about notation. In equation (20) $S_{\mathrm{max}}$ was a "user-defined maximum saturation change", now it is something calculated from the solution...

- line 306-315: If the mixed form of Richards' equation is used, with a (locally mass-conservative) finite volume discretisation and the linear equations are solved sufficiently accurate, why should there be mass balance at all? It is obvious from the beginning, that this could only hint to errors in your code. Thus the statement in line 314-315 is trivial.

- line 328-330: I do not understand, why the computational costs of the time-adaptive algorithm are calculated by $(N_{sol} + N_{param})/2$. For each iteration step in the iterative scheme you have to calculate the nonlinear parameters and their derivatives, assemble a matrix and solve a linear equation system. For each iteration step in the time-adaptive scheme you have to assemble a matrix and solve a linear equation system, while you have to calculate the non-linear parameters and their derivates only once for each time step. So the cost reduction depends on the number of iterations necessary (if it is always one iteration, there is no cost reduction at all) and on the relative computational cost of nonlinear parameter evaluation compared to assembly and solution of the linear equation system. Why should this result just in this simple formula?

- figure 4, 6 and 8: for the two saturation-based schemes which allow the highest precision in all three scenarios, there is often a reduced increase of precision with costs at high precision. This could be a hint that the spatial resolution was not high enough and that in this cases the spatial discretisation error became relevant. I would thus not agree with the conclusions in line 368-371.

- figure 4, 6 and 8: there is something strange with all the figures. While in the tables there are only values for four precisions given, there are always six points in the figures for the truncation based algorithms but only four points for all other algorithms. This does not make sense.

- line 342-348: As both stopping criteria for the non-linear iterations are not very adequate and a condition based on the reduction of non-linear defect should be used, I will not comment on the comparison of this non-adequate criteria.

- line 372-375: I do not agree with the last statement. As the saturation based time stepping TA_S already produced the same precision when a precision of $10^{-4}$ was demanded, it also had a comparable efficiency with the truncation error based algorithm for this case. The only problem was, that the error was not

reduced with the higher precision, probably linked to a not fine enough spatial grid. A not mentioned point is, that for the saturation based time step control, there was a linear decrease of the error with the specified precision, whereas this was more erratic for the truncation based time step control.

- line 398-401: I do not understand this statement. After all, the algorithm did compute a solution, so why was the time step too long for reaching convergence? And if it did no reach convergence, how could it calculate the next time steps?

- line 405-407: Actually, the first two scenarios also had a step change of boundary conditions at the beginning and thus a "non-monotonic" change of boundary conditions. Thus this is not really completely different

- line 410: "to avoid a too rough discretisation of the upper boundary conditions": did you make sure that the times at which the boundary condition changed where reached exactly? If you did not do this, you get unnecessarily wrong solutions. This is not a question of the time stepping strategy, but of common sense and not difficult to implement. As I do not know if this was done, I am not going to discuss the further results of test case 3.

- line 447-449: this also means that most of the algorithms are not really suitable for error control. The relation between specified precision and obtained error is not linear for most of the algorithms.

- line 450-452: This is a trivial remark as a locally mass conservative discretisation scheme is used. It would be different for e.g. standard finite-elements as used in Hydrus.

- line 453-456: What should really be implemented is a convergence condition based on a reduction of the non-linear residual.

- line 457-460: This should be formulated much clearer: The time-adaptive algorithm with the truncation based time-stepping condition did fail to produce accurate results for almost all test cases and converged to the wrong result in the first test cases. Thus it is useless. I would not expect that this will change for 2D or 3D problems. With the saturation-based time-stepping, the time-adaptive algorithm was overall comparable to the standard iterative approach. However, it always was rather costly at high precision, where the time steps are small and thus the number of iterations per time step was also small. Thus the advantage of not calculating the non-linear parameters did not pay off. This also should be similar for 2D and 3D calculations.

- line 462-468: I still do not get, why the time truncation error should be a relevant stopping condition for the non-linear iterations within one time step. Obviously the maximal change of the potential alone is not a reasonable condition, as it is linked to the fluxes and saturation changes via highly non-linear functions.

---

## Author Comment (AC1) · 10 Apr 2017

This manuscript provides a performance assessment of different known algorithms to solve the Richard's equation. In particular, the authors investigate the performance of the Ross method versus the Newton-Raphson with different time-stepping strategies. A nice set of guidelines are provided in the end. The article is well written and provides a nice contribution in this area.

We thank the referee for his/her review whose constructive comments helped in improving the manuscript. We are of course pleased that she/he considers that the manuscript presents a nice contribution to the challenging problem of solving Richards equation.

Based on this, I suggest the publication of the manuscript after minor revision is addressed to tackle this points:

**Minor comments:**

The author should clearly state the assumptions in Equation (1), rigid solid matrix (negligible dro/ro) changes in porosity) but also need to say that $\frac{1}{\rho}\nabla\rho \approx 0$. In this context, it is worth mention that the specific storage coefficient used in Equation (1) is not exactly the same as the specific storage coefficient of the flow equation. The specific storage coefficient is the sum of compressibility of water and soil. In equation (1) the changes in porosity are neglected and therefore "so" is not exactly the specific storage. Only the part corresponding to the compressibility of water.

We will change the text accordingly.

Line 36: actually there are three standard forms of the equation: pressure, saturation and mixed

We do not understand this comment. We wrote L36 'Equation (1) is also called the mixed form of RE. Two alternative formulations exist for RE' and showed the three forms of the RE (eq. 1, 2 and 3).

Equation (13) may be is worth to explain how to calculate fluxes q or simply refer to the appendix here for an example.

It is explained in the appendix, eq A32. We will refer to it.

Equation (15), maybe is worth explaining index k

It is explained two lines later (L147). The reviewer probably refers to equation 13 where we did not explain k. The text will be modified accordingly.

It is not clear whether the method suggested by Ross (2003) is mathematically equivalent to Newton-Raphson or simply performs the same way in this example. In case it is mathematically equivalent, a more detail derivation is required. In case it performs equally in

this case the manuscript should clearly state this fact. Could it be that in2D and 3D the performance of these two algorithms are different?

The method suggests by Ross is mathematically equivalent to Newton Raphson as we explain in equation (8)-(13). We will provide more details.

---

## Author Comment (AC2) · 10 Apr 2017

The paper compares the efficiency of a combination of two linearisation schemes for the solution of the non-linear Richards' equation with different time adaptation criteria. The first scheme is a method presented by Ross in 2003, a kind of semi-implicit scheme, calculating the non-linearities with the solution of the last time step. The second scheme is using Newton iterations. The authors first show, that if applied to the water content-formulation of Richards' equation, the method of Ross is equivalent to the first iteration of a Newton iteration. As the water-content form is only applicable to strictly unsaturated conditions, they use discretisations of the mixed form for the rest of the paper. In the Ross-type scheme, called time-adaptive (though both schemes use adaptive time stepping), the authors apply only the first-iteration of a Newton-scheme, calculating the coefficients again with the old solution, and shorten the time step until convergence. In the Newton-iteration scheme they calculate the coefficients with the last iterate until convergence. Thus in the Ross-type scheme the assembly of the linear-equation system to be solved is faster for the second or later iterates. For the adaption of the time step the authors either use an heuristic approach based on the number of Newton iterations (only for the Newton-based scheme), an approach based on an estimation of the truncation error, or a limit on the maximal allowed change of saturation. The different combinations of time-step control and linearisation approach are applied to three different test cases from the literature. The computational costs, measured in a normalized number of solves, are plotted against precision, measured as the deviation of the results from a reference solution calculated with a very fine time step and a given grid size. The authors conclude that there was no real advantage of the Ross-type scheme.

*General comments:*

The authors address a question, which has been intensively discussed in the last decades. Numerous papers on the best linearisation schemes and time-step adaptation procedures can be found easily in the literature, partially co-authored by one of the authors of this paper, many of them also cited in the paper. Thus the main question is, if the analysis of a very special scheme is a meaningful contribution to the literature and suited for publication in HESS. As there remain a lot of questions to be addressed (see specific comments below), the paper could be accepted only after major revisions. However, I am not convinced that the contributions made by the paper will be significant even after revision.

We fully agree that this question has been intensively discussed. However, we believe that the existing algorithms are still not efficient enough, especially for the recent developments of large scale models used to simulate climate change or to compute global water balances for example. It is more and more recognized that water flow in the unsaturated zone has to be modelled using mechanistic models to improve the reliability of large scale models. However, the difficulty in solving Richards equation in an efficient way (i.e. avoiding time steps in the order of minutes for simulations over several years) hampers its use in large scale models. Therefore, we believe that there is a real need of efficient algorithms for solving Richards equation. We will add a few sentences on this motivation in the introduction.

Did we find THE algorithm? No, unfortunately …

Is our contribution significant? We show for the first time that (i) the 'new' Ross method is a Newton-Raphson method, (ii) the algorithm which performs the best for each test case does not exist amongst the most existing popular algorithms and (iii) two stopping criteria have to be used instead of only one as it is implemented in many codes.

***Specific comments:***

equation (1) + (2): As a rigid solid matrix is assumed, $s0$ could only describe the compressibility of the fluid. As water is nearly incompressible at the pressures occurring in variably saturated soils, the compressibility term is unnecessary and should be dropped.

> We agree with the physical meaning of s0. However, from a numerical point of view, this term is very useful for solving saturated/unsaturated problems in transient.

line 177: "The time-adaptive algorithm consists of keeping the pressure head constant and changing the time step length." Actually, this formulation is misleading. For each tested time step a new solution for the pressure heads is calculated. Thus they are not kept constant. However, the non-linear parameters are always calculated with the solution from the old time step, corresponding to a semi-implicit scheme. Even the matrix has to be reassembled for each tested time step. Thus only the evaluation of the non-linear functions is avoided. An alternative would be the use of an interpolation table for the hydraulic functions to reduce the computational costs and still keep the accuracy high. The misleading formulation is also used in line 6 of the abstract.

> We fully agree. We will improve the text accordingly.

line 210: I do not understand this formulation $\max_i = \left(\left|\Delta S_{max}\right|\right)$ is the maximum of the actual change, how can it exceed itself? Do you mean exceeds $(1+\lambda)\Delta S$max?

> The equation did not appear properly in the manuscript provided by HESS, sorry.
>
> $\Delta t^{n+1}$ is an estimate of the next time step. After computation over $\Delta t^{n+1}$, the saturation change can exceed the user provided $S_{max}$. We will provide some more details in the revised version.

line 219-226: Is this important here? If necessary at all, please move it to the introduction

> We think we have to refer to this kind of approach and we will move this part in the introduction.

line 243: replace "superior to" by "larger than"

> Will be changed in the revised version.

line 253: "Implicit standard finite volumes" is not really a precise description. I guess you mean a cell-centred finite volume scheme for the spatial discretisation with an implicit Euler-scheme for the temporal discretisation. Actually, already in chapter 3, equation (15) the discretisation is given. Shouldn't you just refer to that section?

You are right concerning the method we used and we will provide the detailed information. However, equation (15) is more general. It also holds for other spatial discretizations like finite elements.

line 261: "the error based on the maximum change of the state variables between two iterations" would be $\max_i = \left| \psi_i^{n+1,k+1} - \psi_i^{n+1,k} \right|$. If your formula is correct you are looking at "the error base on the maximal change of the state variables in the last iteration". This actually is a very bad convergence condition as it cannot distinguish between "already converged" and "no convergence at all". However, it is also completely unclear to me, why the time truncation error should be a sensible stopping condition. A reasonable stopping condition is based on the reduction of the non-linear residual compared to the initial non-linear residual. This would really be related to a reduction of the error in the solution of the non-linear equation.

Concerning the first criterion, it is not in the last iteration (see iteration numbered by k). It is during the iterative process. If this criterion is met, the process is stopped and the computation of the next time step is performed. This is a very popular stopping method, not only for unsaturated flow but also for density driven flow for example.

Time stopping criteria have been applied by others (see references in the manuscript).

Residuals are also used as stopping criterion but it performs like the criterion based on the maximum change of the state variable. Both criteria are linearly linked (Ackerer et al. 1999 Modeling Variable Density Flow and Solute Transport in Porous Medium: 1. Numerical Model and Verification. Transport in Porous Media 35: 345–373).

We will add the following in the revised manuscript:

We checked the error on the residuals for the two kinds of stopping criteria (pressure based and truncation). The maximum absolute value of the residual was always smaller than the error used in the stopping criterion. Our stopping criteria are more restrictive than the absolute residual error.

equation (27) and (28): is it really necessary to write out this equations? Is it not enough to state that relative and absolute error bounds are given?

We will change the text accordingly.

line 269-271: Actually, not all possible combinations have been performed. You could also have tested using only the truncation error (if this makes sense).

We make all possible combinations but we reported only the most significant results in the paper as stated in L270-L271.

line 293-294: as the spatial discretisation error (though not explicitly mentioned) is addressed here: How did you check, that the grid really was fine enough? As you try to get very accurate solutions in time (down to an error of $10^{-5}$), did you really make sure, that the grid is fine enough to produce changes significantly lower than $10^{-5}$ if further refined?

We did it in the traditional way, by successive grid refinements. See below, discussion on figures 4,6,8…

line 301-303: As you are using a mixed scheme: why did you not just calculate $\Delta S\max$ from the saturations? I am also a bit confused about notation. In equation (20) $S\max$ was a "user-defined maximum saturation change", now it is something calculated from the solution...

We want to compare methods that use different criteria to stop the iterative procedure. Equation (29) gives the relationship between both user's defined criteria.

We will better explain it in the revised version.

line 306-315: If the mixed form of Richards' equation is used, with a (locally mass-conservative) finite volume discretisation and the linear equations are solved sufficiently accurate, why should there be mass balance at all? It is obvious from the beginning that this could only hint to errors in your code. Thus the statement in line 314-315 is trivial.

We agree. We mentioned that the mass balance errors were negligible and our comparisons are not based on this error. We just check this error because it is commonly used to compare numerical schemes (even the codes which solve mixed form of RE).

line 328-330: I do not understand, why the computational costs of the time-adaptive algorithm are calculated by (*Nsol+Nparam*)/2. For each iteration step in the iterative scheme you have to calculate the nonlinear parameters and their derivatives, assemble a matrix and solve a linear equation system. For each iteration step in the time-adaptive scheme you have to assemble a matrix and solve a linear equation system, while you have to calculate the non-linear parameters and their derivates only once for each time step. So the cost reduction depends on the number of iterations necessary (if it is always one iteration, there is no cost reduction at all) and on the relative computational cost of nonlinear parameter evaluation compared to assembly and solution of the linear equation system. Why should this result just in this simple formula?

We assume that the computational costs are depending on the time required to compute the non-linear parameters and the time required to solve the system of equations i.e. Nparam, the number of calls of the subroutine which computes the parameter values and Nsol the number of calls to solve the equations. For the standard approach, Nparam is equal to Nsol. So we have computational costs that are equal to 2Nsol for the standard approach and (Nsol+Nparam) for the time adaptive scheme. This is why we used Nsol for the standard approach and (Nsol+Nparam)/2 for the time adaptive scheme.

figure 4, 6 and 8: for the two saturation-based schemes which allow the highest precision in all three scenarios, there is often a reduced increase of precision with costs at high precision. This could be a hint that the spatial resolution was not high enough and that in this cases the spatial discretisation error became relevant. I would thus not agree with the conclusions in line 368-371.

We agree on the reduced increase of precision and we do not have any clear explanation. The grid size is the same for all schemes, so we assume that the error due to the spatial discretization is the same for all schemes.

We redo the computation of TC1 with a spatial discretization two times finer and relative tolerances of $10^{-4}$ and $10^{-5}$. The differences could not be seen on the profile (see figure below obtained for a tolerance of $10^{-5}$). Therefore, we disagree with the reviewer's statement; the spatial discretization is high enough.

[Figure]

figure 4, 6 and 8: there is something strange with all the figures. While in the tables there are only values for four precisions given, there are always six points in the figures for the truncation based algorithms but only four points for all other algorithms. This does not make sense.

> Tables with too many numbers are boring. We provide more point in figures to better underline the trends which are difficult to see in tables.

line 342-348: As both stopping criteria for the non-linear iterations are not very adequate and a condition based on the reduction of non-linear defect should be used, I will not comment on the comparison of this non-adequate criteria.

> The criteria are quite popular and, for our examples, more restrictive than residual based stopping criteria. We will develop this in the revised manuscript.

line 372-375: I do not agree with the last statement. As the saturation based time stepping TA_S already produced the same precision when a precision of $10^{-4}$ was demanded, it also had a comparable efficiency with the truncation error based algorithm for this case. The only problem was, that the error was not reduced with the higher precision, probably linked to a not fine enough spatial grid. A not mentioned point is, that for the saturation based time step control, there was a linear decrease of the error with the specified precision, whereas this was more erratic for the truncation based time step control.

> As already mentioned, it is not a problem of spatial discretization.
> We will mention the linear decrease in the revised version.

line 398-401: I do not understand this statement. After all, the algorithm did compute a solution, so why was the time step too long for reaching convergence? And if it did not reach convergence, how could it calculate the next time steps?

> We revised figure 7 and changed the text accordingly. There was some mixed up use of the data files. We checked all computations and dataset. Thanks for pointing out this error.

line 405-407: Actually, the first two scenarios also had a step change of boundary conditions at

the beginning and thus a "non-monotonic" change of boundary conditions. Thus this is not really completely different

> Table 2 clearly indicates that the boundary conditions are not changing in time for the first two scenarios.

line 410: "to avoid a too rough discretisation of the upper boundary conditions": did you make sure that the times at which the boundary condition changed where reached exactly? If you did not do this, you get unnecessarily wrong solutions. This is not a question of the time stepping strategy, but of common sense and not difficult to implement. As I do not know if this was done, I am not going to discuss the further results of test case 3.

> The upper boundary fluxes change with a time step of 1 day (see Fig. 9). To well describe this time varying boundary conditions, we fixed the maximum time step length to 0.2 day.

> For a given time, the boundary conditions are linearly interpolated. We will explain this issue more properly in the revised manuscript.

line 447-449: this also means that most of the algorithms are not really suitable for error control. The relation between specified precision and obtained error is not linear for most of the algorithms.

> We agree.

line 450-452: This is a trivial remark as a locally mass conservative discretisation scheme is used. It would be different for e.g. standard finite-elements as used in Hydrus.

> It is trivial for the experts and it would not be different for finite elements schemes which also preserve mass if the mass balance is computed consistently with the method i.e. on the dual mesh.

line 453-456: What should really be implemented is a convergence condition based on a reduction of the non-linear residual.

> As stated previously, the stopping criteria we used are more restrictive.

line 457-460: This should be formulated much clearer: The time-adaptive algorithm with the truncation based time-stepping condition did fail to produce accurate results for almost all test cases and converged to the wrong result in the first test cases. Thus it is useless. I would not expect that this will change for 2D or 3D problems. With the saturation-based time-stepping, the time-adaptive algorithm was overall comparable to the standard iterative approach. However, it always was rather costly at high precision, where the time steps are small and thus the number of iterations per time step was also small. Thus the advantage of not calculating the non-linear parameters did not pay off. This also should be similar for 2D and 3D calculations.

> Time adaptive performed quite well for the test TC2 (see fig. 6). The difference between our 1D and 2-3D calculations is of course the number of elements and therefore the number of time the parameters have to be computed. Therefore, for a given accuracy, the time adaptive algorithm might be more efficient. We will reformulate this part.

line 462-468: I still do not get, why the time truncation error should be a relevant stopping condition for the non-linear iterations within one time step. Obviously the maximal change of the potential alone is not a reasonable condition, as it is linked to the fluxes and saturation changes via highly non-linear functions.

Our study shows that the time truncation error is a relevant criterion. The three test cases that we have done show the relevance of the algorithm that uses the time truncation error and the maximum change of the pressure. The reason is twofold:

- Non linearity which is controlled by the maximum change of a variable between two successive iterations. We choose here the maximum change of the state variables (pressure or saturation) which is more restrictive than residual errors.

- The approximation of the derivative in time which is first order. It is possible to fulfill the criterion related to the iteration but still have a poor approximation of the time derivative due to a too large time step. Therefore, we checked the truncation error.

This is why both criteria are useful for solving the Richards equation.